# R1441C and G2019S LRRK2 knockin mice have distinct striatal molecular, physiological, and behavioral alterations

Harry S. Xenias[1], Chuyu Chen [2], Shuo Kang[2], Suraj Cherian[1], Xiaolei Situ[2], Bharanidharan Shanmugasundaram[2], Guoxiang Liu[3], Giuseppe Scesa[2], C. Savio Chan [1✉] & Loukia Parisiadou [2✉]

LRRK2 mutations are closely associated with Parkinson's disease (PD). Convergent evidence suggests that LRRK2 regulates striatal function. Here, by using knock-in mouse lines expressing the two most common LRRK2 pathogenic mutations—G2019S and R1441C—we investigated how LRRK2 mutations altered striatal physiology. While we found that both R1441C and G2019S mice displayed reduced nigrostriatal dopamine release, hypoexcitability in indirect-pathway striatal projection neurons, and alterations associated with an impaired striatal-dependent motor learning were observed only in the R1441C mice. We also showed that increased synaptic PKA activities in the R1441C and not G2019S mice underlie the specific alterations in motor learning deficits in the R1441C mice. In summary, our data argue that LRRK2 mutations' impact on the striatum cannot be simply generalized. Instead, alterations in electrochemical, electrophysiological, molecular, and behavioral levels were distinct between LRRK2 mutations. Our findings offer mechanistic insights for devising and optimizing treatment strategies for PD patients.

[1] Department of Neuroscience, Feinberg School of Medicine, Northwestern University, Chicago, IL, USA. [2] Department of Pharmacology, Feinberg School of Medicine, Northwestern University, Chicago, IL, USA. [3] Department of Neurology, Feinberg School of Medicine, Northwestern University, Chicago, IL, USA. ✉email: saviochan@gmail.com; loukia.parisiadou@northwestern.edu

## Main

T he identification of LRRK2 mutations provided important insights into the genetic basis of Parkinson's disease (PD). First, autosomal dominant mutations in LRRK2 are the most common genetic cause of late-onset PD[1,2]. Second, compelling evidence from genome-wide association studies has identified *LRRK2* as a risk factor for sporadic PD[3–5]. Patients with LRRK2 mutations exhibit clinical and pathological phenotypes indistinguishable from sporadic PD[6–8], suggesting common disease mechanisms. These findings urge further efforts to understand mutant LRRK2 pathophysiology. The gained knowledge about the impact of different LRRK2 mutations can be leveraged for therapeutics in PD.

The LRRK2 mutations G2019S and R1441C (hereafter referred to as GS or RC) are commonly found in familial PD[9–11]. These two mutations are located in the kinase and ROC (Ras of complex GTPase) domains of the LRRK2 protein, respectively (Fig. 1a)[12]. Although it is widely accepted that all pathogenic mutations increase LRRK2 kinase activity[13,14], the mechanism through which LRRK2 leads to the disease neuropathology remains unclear. It is also still unknown whether the different pathogenic mutations alter the biochemical properties of the LRRK2 protein through common or distinct mechanisms. The literature focuses on the GS mutation; however, we lack systematic comparisons across mutations to define a coherent framework of LRRK2-mediated dysfunction. Relatedly, previous studies support distinct clinical features associated with different *LRRK2* variants[15,16], likely reflecting a mutation specific dysfunction.

Several transgenic and gene-targeted knock-in (KI) mice expressing LRRK2 GS and RC mutations have hinted at a critical role of LRRK2 in regulating dopamine release in the striatum[17–20]. However, as the results are highly inconsistent across studies, additional efforts are needed to clarify the precise alterations induced by LRRK2 mutations in nigrostriatal dopamine transmission. LRRK2 levels are high in striatal projection neurons (SPNs)[21,22]; accumulating evidence shows that LRRK2 regulates synaptic events in SPNs[23–25]. While similar alterations in synaptic functions were observed in SPNs with RC and GS mutations, distinct molecular and physiological perturbations unique to the RC mutation were reported. These include elevated PKA signaling in the striatum of RC and not GS mice and increased synaptic glutamate expression in SPNs of the RC KI mice[26]. Accordingly, emerging evidence suggests that LRRK2 phosphorylation is differentially affected by LRRK2 disease-associated variants implicating distinct phosphoproteome signatures across LRRK2 mutations[27].

Overall, these findings of divergent effects between RC and GS mutations highlight the importance of more detailed comparative studies to define the molecular underpinnings and functional dysregulations with LRRK2 mutations. By using gene-targeted KI mice, we systematically investigated the alterations in striatal electrochemical, electrophysiological, molecular, and motor functions associated with LRRK2 RC and GS mutations. Our results showed alterations unique to specific LRRK2 mutations.

## Results

### R1441C and G2019S LRRK2 mice have decreased nigrostriatal dopaminergic transmission

Studies that examine dopamine transmission in LRRK2 mutant mice have not yielded consistent results[17,28–31]. This could be partly due to the employment of BAC (bacterial artificial chromosomes) or other transgenic lines confounded by unintended genomic alterations in expression patterns and levels of endogenous LRRK2[32,33]. In contrast, KI mouse models serve as a well-validated approach for studying LRRK2 mutations in relevant cell types at the physiologically relevant expression level[17,26]. In this study, we used adult LRRK2

RC and GS KI mice that express mutant LRRK2 proteins under the regulation of the endogenous promoter (Fig. 1a). We first examined nigrostriatal dopamine transmission with fast-scanning cyclic voltammetry in striatal tissues from WT, RC, and GS mice (Fig. 1b). We found a decrease in evoked dopamine release in both RC and GS mice compared to WT mice: $[DA]_{WT} = 1057.6 \pm 117.9$ nM ($n = 10$ mice); $[DA]_{RC} = 538.7 \pm 143.2$ nM ($n = 9$ mice; $p = 0.006$, Mann–Whitney $U$ test); $[DA]_{GS} = 554.9 \pm 218.4$ nM ($n = 13$ mice, $p = 0.001$; Mann–Whitney $U$ test). There were no differences in evoked dopamine release between RC and GS mice ($p = 0.69$, Mann–Whitney $U$ test) (Fig. 1c, d) and no sex effect. See Supplementary Table 1 for a statistical summary. Codes are openly available at Zenodo.

### iSPNs have decreased excitability in R1441C LRRK2 mice

It is established that dopamine modulates the excitability of SPNs[34–38]. Accordingly, the disruption of dopamine signaling profoundly alters the intrinsic properties of SPNs[39,40]. Given our finding that both RC and GS mice had decreased dopamine release, we sought to examine if direct- and indirect-pathway SPNs (dSPNs and iSPNs) have altered excitability in the LRRK2 mutant mice. Whole-cell, current-clamp recordings were performed on identified SPNs in WT, RC, and GS mice (Fig. 2).

We found no differences in the number of spikes in dSPNs between WT and RC mice for the current (I) that elicited the half-maximum firing from dPSNs in the WT mice (I = 475 pA, $output_{WT} = 12.0 \pm 4.0$ spikes, $n = 20$ cells; $output_{RC} = 14.0 \pm 3.0$ spikes, $n = 30$ cells; $p = 0.21$, Mann–Whitney $U$ test) (Fig. 2a). In contrast, there was a decrease in the number of evoked spikes for the iSPNs in RC mice for the current that yielded the half-maximum firing from iSPNs in the WT mice (I = 325 pA, $output_{WT} = 14.0 \pm 5.0$ spikes, $n = 20$ cells; $output_{RC} = 6.0 \pm 6.0$ spikes, $n = 23$ cells; $p = 0.037$; Mann–Whitney $U$ test) (Fig. 2b). The individual data points are shown in Supplementary Fig. 1. To quantify the decreased excitability in the iSPNs of RC mice, we compared input-output functions for the WT and RC mice. As shown in Fig. 2b, the maximal difference in the spike output was observed at I = 625 pA; a lower number of spikes were elicited in iSPNs of RC mice compared to WT mice ($output_{WT} = 22.0 \pm 4.5$ spikes, $n = 17$ cells; $output_{RC} = 15.0 \pm 5.0$ spikes, $n = 23$ cells; $p = 0.0010$, Mann-Whitney $U$ test) (Fig. 2b). In contrast, we found no detectable differences in GS mice as measured with the half-maximum firing responses for either dSPNs ($output_{WT} = 8.0 \pm 5.0$ spikes, $n = 13$ cells; $output_{GS} = 11.0 \pm 3.0$ spikes, $n = 29$ cells; $p = 0.12$, Mann-Whitney $U$ test) or iSPNs ($output_{WT} = 10.5 \pm 4.8$ spikes, $n = 20$ cells; $output_{GS} = 12.0 \pm 4.0$ spikes, $n = 28$ cells, $p = 0.73$, Mann-Whitney $U$ test) (Fig. 2c, d). See Supplementary Table 2 for a statistical summary of the half-maximum and maximal F-I differences in firing.

We further analyzed the maximum firing across the current injection steps per cell. In agreement with the above analysis, we found reduced maximal firing only in RC iSPN ($output_{WT} = 23.0 \pm 4.5$ spikes, $n = 16$ cells; $output_{RC} = 18.5 \pm 3.5$ spikes, $n = 16$ cells; $p = 0.0114$, Mann-Whitney $U$ test). There were no significant differences between WT and RC iSPNs for the injection currents generating maximal firing ($current_{WT} = 525.0 \pm 125$ pA, $n = 16$ cells; $current_{RC} = 500.0 \pm 100$ pA, $n = 16$ cells; $p = 0.46$, Mann-Whitney $U$ test; see Supplementary Fig. 2 and Supplementary Table 3 for statistical summary).

As the excitability of SPNs is a function of dendritic structure[41], we asked whether an increase in membrane capacitance (Cm) could account for the decreased excitability seen in the iSPNs of RC mice. We found an increased membrane capacitance for iSPNs in RC mice compared to WT mice ($Cm_{WT} = 68.8 \pm 13.8$ pF, $n = 27$ cells; $Cm_{RC} = 87.3 \pm 13.3$ pF,

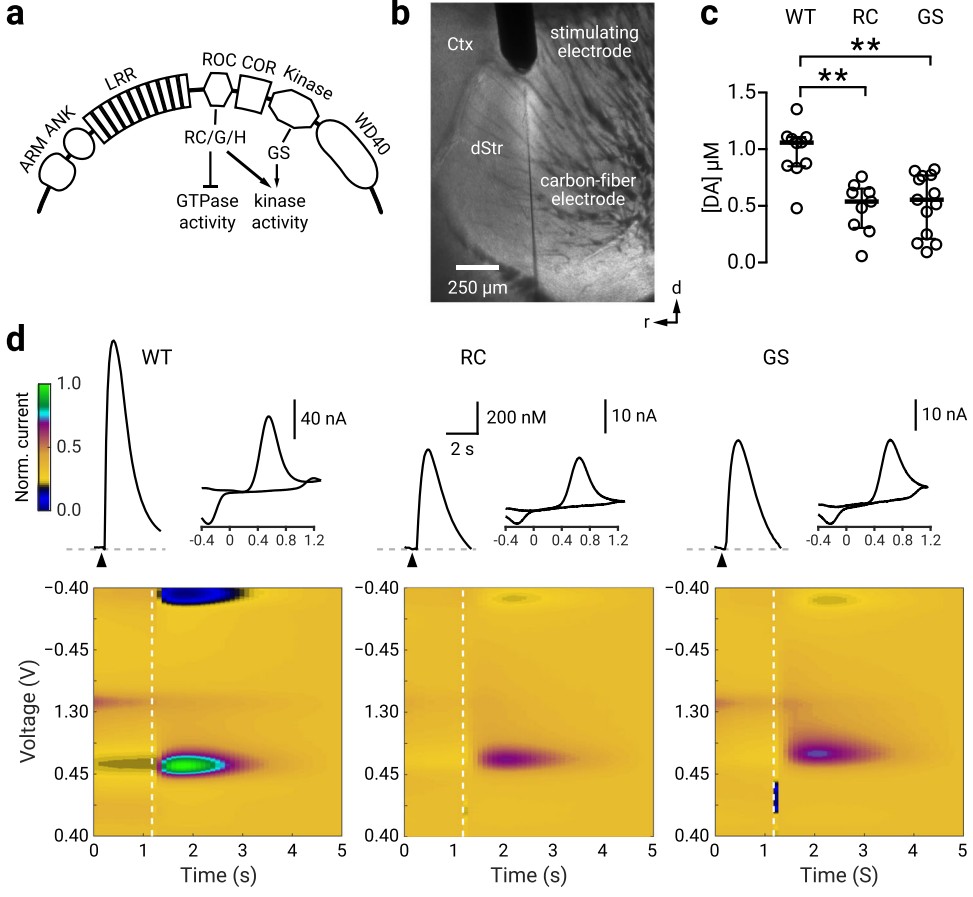

**Fig. 1 RC and GS LRRK2 mice have decreased nigrostriatal dopamine release. a** A schematic diagram of LRRK2 protein: a family of Ras-like G-proteins with functionally distinct multi-domains, consisting of armadillo repeats (ARM), ankyrin repeats (ANK), the leucine-rich repeats (LRR), the C-terminal of Roc (COR), the ROC GTPase domain where the R1441C/G/H (RC) mutations reside, and the kinase domain where the G2019S (GS) mutation resides. The WD40 domain is involved in membrane binding with ARM and ANK believed to stabilize the electrostatic surfaces of the domains. **b**. Brightfield photomicrograph of a parasagittal slice with a concentric stimulating electrode placed in the dorsal striatum (dStr); a carbon fiber electrode was inserted adjacent to the stimulation site. Ctx, cortex. **c** Population data showing for evoked [DA] in WT, RC, and GS LRRK2 KI mice. The median evoked concentrations were: $[DA]_{WT} = 1057.6 \pm 117.9$ nM ($n = 10$); $[DA]_{RC} = 538.7 \pm 143.2$ nM ($n = 9$); $[DA]_{GS} = 554.9 \pm 218.4$ nM ($n = 13$). Compared to WT mice, evoked dopamine [DA] was decreased in RC mice ($p = 0.006$, Mann–Whitney $U$ test). Similarly, compared to WT mice, [DA] was less in GS mice ($p = 0.001$; Mann–Whitney $U$ test). There was no difference in evoked release between RC and GS mice ($p = 0.69$; Mann–Whitney $U$ test). **d** Representative time courses for [DA] with corresponding cyclic voltammograms and color maps of fast-scanning cyclic voltammetry recordings. Peak evoked responses: $[DA]_{WT} = 1223.7$ nM; $[DA]_{RC} = 655.1$ nM; $[DA]_{GS} = 710.4$ nM. Colormaps are normalized to the oxidation current from the WT recording. ** denotes $p < 0.01$. See Supplementary Table 1 for complete sample sizes by sex and statistical results.

$n = 27$ cells; $p = 0.0388$, Mann–Whitney $U$ test). In contrast, no change in Cm was observed for dSPNs between RC and WT mice ($Cm_{WT} = 78.3 \pm 15.5$ pF, $n = 15$ cells; $Cm_{RC} = 81.2 \pm 14.4$ pF, $n = 30$ cells; $p = 0.23$, Mann-Whitney $U$ Test). We also examined for changes in Cm in GS mice. In line with previous findings[25,42], we found no changes in the intrinsic properties of SPNs in the GS mice. We found no differences in membrane capacitance between GS mutant or WT mice for dSPNs ($Cm_{WT} = 88.1 \pm 10.8$ pF, $n = 15$ cells; $Cm_{GS} = 95.3 \pm 12.0$ pF, $n = 27$ cells; $p = 0.28$, Mann-Whitney $U$ Test) or iSPNs ($Cm_{WT} = 74.6 \pm 12.2$ pF, $n = 19$ cells; $Cm_{GS} = 82.6 \pm 15.2$ pF, $n = 30$ cells; $p = 0.50$, Mann-Whitney $U$ Test) (See Supplementary Fig. 3 and Supplementary Table 4 for a complete listing of general membrane electrophysiological properties). Additionally, we examined the action potential properties of the SPNs in both RC and GS mice. In keeping with the hypoexcitability in firing of iSPNs in RC mice, we found that the interspike interval (ISI) of RC iSPNs were significantly increased ($ISI_{WT} = 8.0 \pm 1.0$ ms, $n = 17$ cells; $ISI_{RC} = 9.4 \pm 1.2$ ms, $n = 19$ cells; $p = 0.0388$, Mann–Whitney $U$ Test; Supplementary Fig. 4). These and other general membrane properties and

actional potential characterizations such as the full width at half maximum and rate of membrane potential change (dV/dt) are summarized in Supplementary Tables 4 and 5.

The hypoexcitability of iSPNs in models of PD has been suggested as a homeostatic response to an increased corticostriatal transmission[24,25,43], we thus examined possible presynaptic changes in corticostriatal transmission by measuring the paired-pulse ratios (PPRs) of the corticostriatal responses in WT and RC mice (see Methods). There were no differences in the PPRs for either dSPNs ($PPR_{WT} = 1.31 \pm 0.10$, $n = 14$ cells; $PPR_{RC} = 1.33 \pm 0.15$, $n = 17$ cells; $p = 0.18$, Mann-Whitney $U$ test) or iSPNs ($PPR_{WT} = 1.28 \pm 0.14$, $n = 13$ cells; $PPR_{RC} = 1.54 \pm 0.25$, $n = 18$ cells; $p = 0.11$, Mann-Whitney $U$ test) (Supplementary Fig. 5). Unexpectedly, we found an increase in the 95–5% EPSC decay time of iSPNs in RC mice compared to WT mice ($decay_{WT} = 61.1 \pm 38.7$ ms, $n = 13$ cells; $decay_{RC} = 139.5 \pm 81.1$ ms, $n = 18$ cells; $p = 0.037$, Mann-Whitney $U$ test) (Supplementary Fig. 5). In contrast, there was no difference in EPSC decay times for dSPNs between WT mice and RC mice ($decay_{WT} = 88.7 \pm 40.3$ ms, $n = 14$ cells; $decay_{RC} = 182.9 \pm 136.0$ ms, $n = 17$ cells;

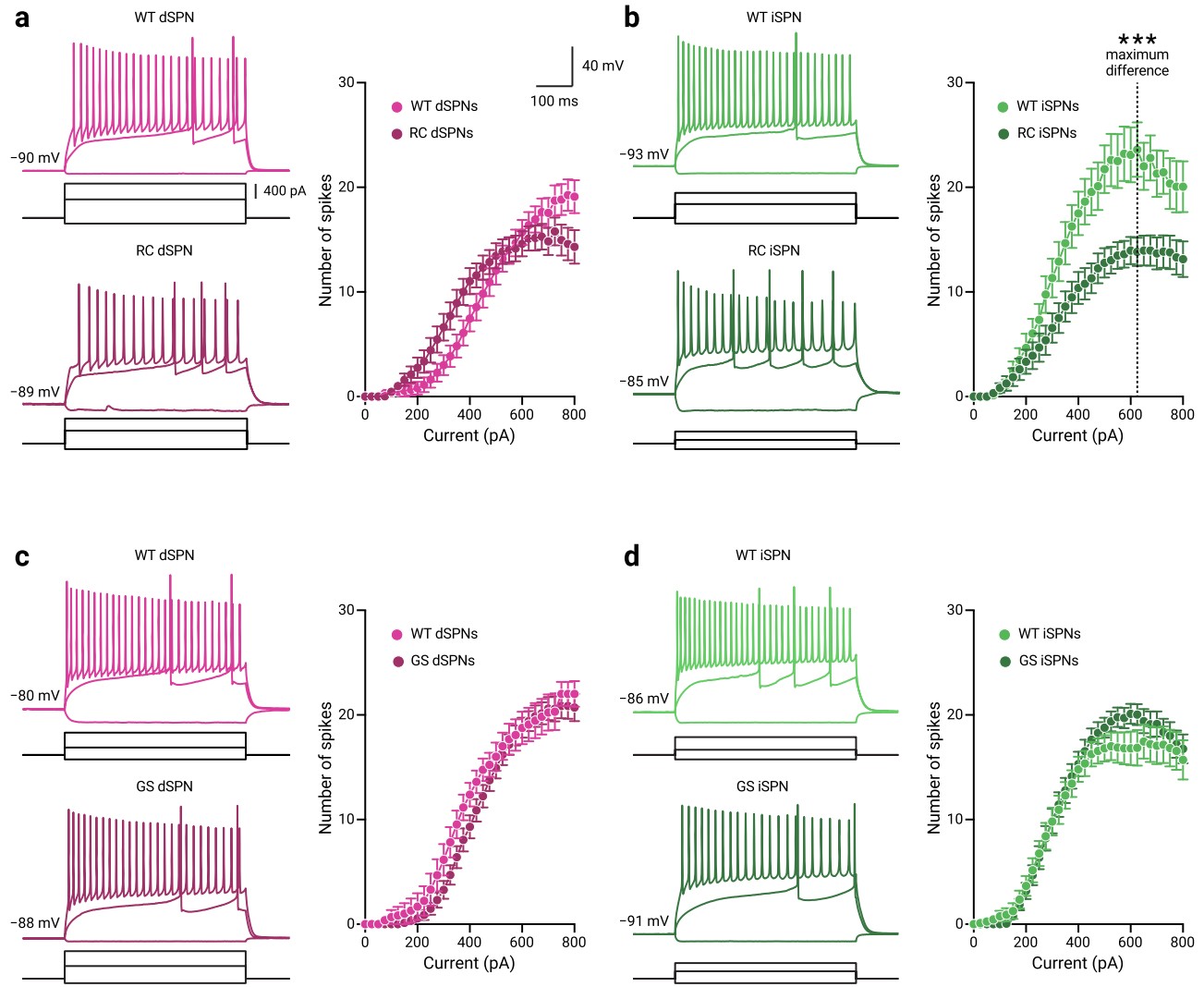

**Fig. 2 iSPNs in RC LRRK2 mice have decreased excitability. a**, **b** Representative whole-cell current-clamp recordings of dSPNs and iSPNs in WT and RC LRRK2 KI mice. The three injection steps shown represent the currents for the different voltage responses, the hyperpolarization response, the first occurrence of action potentials, and the maximum firing, respectively. No differences in the frequency-current (F-I) function were found for dSPNs between WT and RC mice at the current where the half-maximum firing occurred in the population data (I = 475 pA, spikes$_{WT}$ = 12.0 ± 4.0, n = 20 cells; spikes$_{RC}$ = 14.0 ± 3.0, n = 30 cells; p = 0.21, Mann–Whitney U test). For iSPNs there was a decrease of the firing response to half-maximum firing current in the RC mice compared to the WT mice (I = 325 pA, spikes$_{WT}$ = 14.0 ± 5.0, n = 20 cells; spikes$_{RC}$ = 6.0 ± 6.0, n = 23 cells; p = 0.037, Mann–Whitney U test). The current where the maximal difference between the F-I functions for the WT and RC mice was also calculated (I = 625 pA, dotted line) to compare the number of elicited spikes between the WT and RC mice. We found a decrease in the number of spikes elicited in RC mice compared to WT mice (spikes$_{WT}$ = 22.0 ± 4.5, n = 17 cells; spikes$_{RC}$ = 15.0 ± 5.0, n = 23 cells; p = 0.0011, Mann–Whitney U test). **c**, **d** Representative whole-cell current-clamp recordings of dSPNs and iSPNs in WT and GS LRRK2 KI mice with population F-I data. For GS mice, no differences were found in the half-maximum firing responses for either dSPNs (spikes$_{WT}$ = 8.0 ± 5.0, n = 13 cells; spikes$_{GS}$ = 11.0 ± 3.0, n = 29 cells; p = 0.12, Mann–Whitney U test) or iSPNs (spikes$_{WT}$ = 10.5 ± 4.8, n = 20 cells; spikes$_{GS}$ = 12.0 ± 4.0, n = 28 cells; p = 0.73, Mann–Whitney U test). F-I data are shown as mean ± standard error of the mean. * denotes p < 0.05. See Supplementary Table 2 for complete sample sizes and statistical results.

p = 0.13, Mann-Whitney U test). See Supplementary Table 6 for a statistical summary.

**R1441C mice show impaired dopamine-dependent motor learning**. The dorsal striatum plays a critical role in habit and motor learning[44–47]. Recent data show that dopamine loss through receptor antagonism during the acquisition of a motor skill impedes future motor performance even after the cessation of the blockade[48,49]. Given the importance of dopamine dependence on motor learning[48,50] and the reduction of dopamine release in the dorsal striatum in RC and GS KI mice (Fig. 1), we investigated how receptor-specific disruption of dopamine signaling would affect striatal motor learning. We used an

accelerating rotarod task—a well-established paradigm for assessing dopamine-dependent motor learning[48,51–53]. Specifically, we first systemically injected D1 and D2 dopamine receptor antagonists (eticlopride and SCH23390, respectively) separately or in combination (referred to as an 'antagonistic cocktail') during the initial acquisition of the skill to determine the roles of D1 and D2 dopamine receptor-mediated signaling on motor learning in RC and GS mice (Fig. 3a).

Consistent with previous studies that RC KI mice display no overt abnormalities in striatum-dependent motor learning under basal conditions[19,54], we found that saline treated RC mice learned equally well on the rotarod task (Fig. 3b). In addition, our data were consistent with prior reports[48,51] that WT mice treated

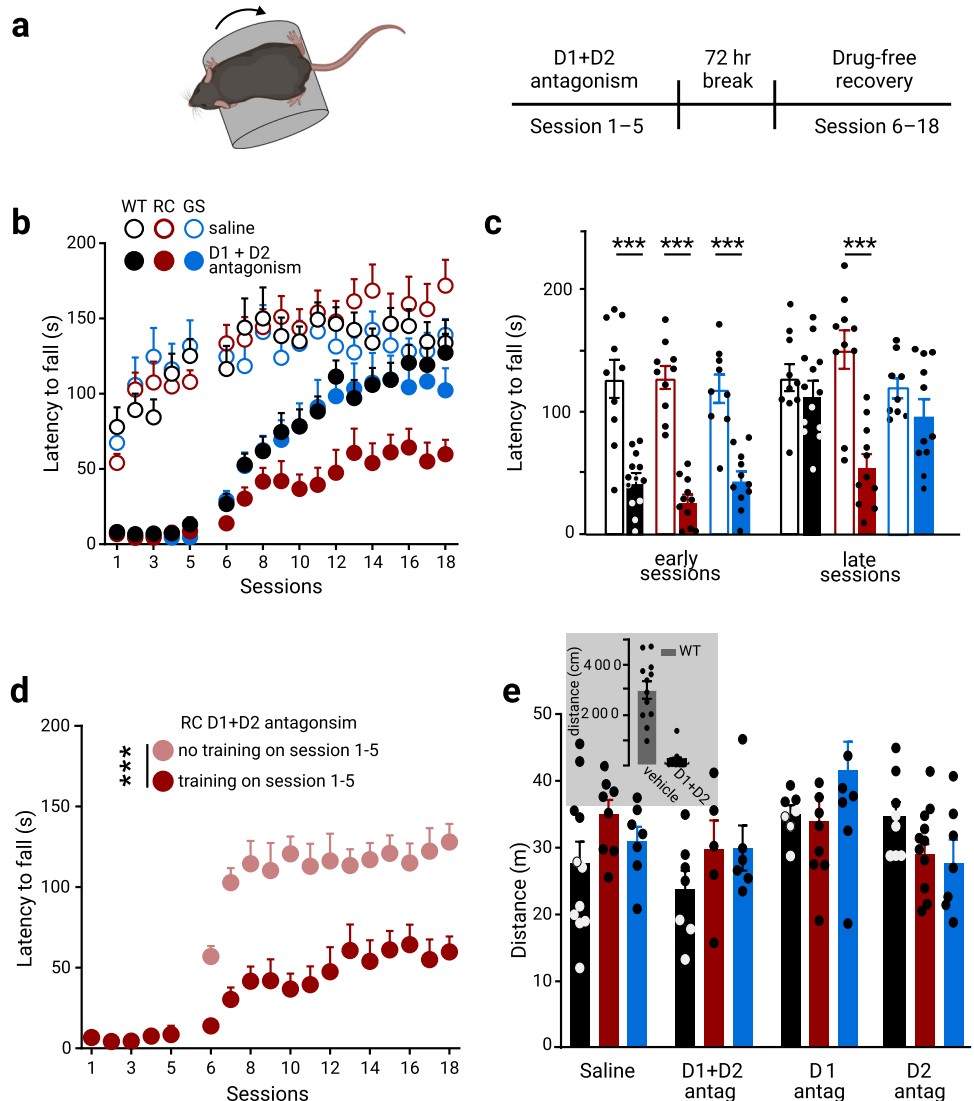

**Fig. 3 RC KI mice show dopamine-dependent motor learning impairments. a** Schematic of the rotarod training paradigm. Mice were assessed over a total of 18 daily sessions, where every daily session consisted of five trials. WT, GS, and RC mice were administered either saline or a cocktail of D1 receptor (SCH23390) and D2 receptor (eticlopride), 1 mg/kg of each antagonist 30 min prior to daily sessions, and trained for five successive days in an accelerated rotarod. After a 72 h break, the mice were returned to the rotarod for an additional 13 days of a drug-free recovery phase. **b** Genotype-dependent effect of blocking both D1 and D2 receptors during the first five sessions on rotarod performance of WT, RC, and GS mice. The average latency in the early (session 6–8) and late (session 16–18) stages in the drug-free recovery phase in **b** is summarized in **c**. Open and filled bars represent the average performance of saline control and drug-treated mice, respectively (saline treated groups: $n_{WT} = 10$ mice, $n_{RC} = 10$ mice, and $n_{GS} = 9$ mice; D1 + D2 receptor antagonist treated groups: $n_{WT} = 13$ mice, $n_{RC} = 11$ mice, and $n_{GS} = 11$ mice; ***$p < 0.001$ vs genotype-matched saline control). **d** Improved performance of RC mice ($n = 10$) received antagonist cocktail but no training during the first five sessions. The RC$_{D1+D2}$ group data is from Fig. 3b–c for reference. See Supplementary Table 7 for complete sample sizes and statistical results. **e**. Distance traveled in the open field after five days of rotarod training with either vehicle or dopamine receptor antagonist cocktail treatment. Another group of mice received dopamine antagonist treatment and rotarod training for five days under the same schedule described in a-c and were tested in the open field immediately after the 72-hr break instead of the drug-free phase (saline treated groups: $n_{WT} = 11$ mice, $n_{RC} = 8$, and $n_{GS} = 7$ mice; D1R + D2R antagonist treated groups: $n_{WT} = 7$, $n_{RC} = 5$, and $n_{GS} = 6$ mice; D1R antagonist treated groups: $n_{WT} = 7$, $n_{RC} = 8$, and $n_{GS} = 9$ mice; D2R antagonist treated groups: $n_{WT} = 8$, $n_{RC} = 11$, and $n_{GS} = 7$ mice).

with antagonists' cocktail during the initial five-day training period (session 1–5; Fig. 3a) have dramatic impairments in the rotarod performance, regardless of genotype. Here, we showed that upon 72 h of recovery from the last antagonist administration, the latency to fall for WT, RC, and GS mice although initially degraded (compared to saline treated controls), (Fig. 3b), increased over subsequent sessions in different manners (saline: $n_{WT} = 10$ mice, $n_{RC} = 10$ mice, and $n_{GS} = 9$ mice; D1 + D2 receptor antagonists: $n_{WT} = 13$ mice, $n_{RC} = 11$ mice, and $n_{GS} = 11$ mice; treatment × genotype × session interaction, $F_{102, 1887} =$

1.95, $p < 0.001$, 3-way ANOVA with repeated measures) (Fig. 3b). The initial impaired motor performance of the drug-free phase of the task was only gradually improved towards the end of the task in WT mice[51]. Thus, we focused on comparing the performance of the mice across genotypes over the last sessions. The average latency to fall in the drug-treated WT and GS mice showed no difference compared to saline-treated controls over the training sessions (session 16–18; $p = 1.0$, latency$_{WT\_saline} = 137.2 \pm 11.9$ s, latency$_{WT\_antagonists} = 124.3 \pm 10.9$ s; $p = 0.98$, latency$_{GS\_saline} = 130.9 \pm 8.7$ s, latency$_{GS\_antagonists} = 104.9 \pm$

14.1 s, 3-way ANOVA, repeated measures) (Fig. 3b, c, Supplementary Table 7). In contrast, dopamine receptor antagonism unmasked a deficit in RC KI mice; they failed to reach the performance of their corresponding saline-treated controls over the training sessions (trial × genotype × drug interaction: $F_{6,116} = 2.24$, $p = 0.044$; $latency_{RC\_saline} = 162.4 \pm 17.0$ s, $latency_{RC\_antagonists} = 59.7 \pm 11.0$ s, $p < 0.001$, 3-way ANOVA with Tukey *post hoc* test) (Fig. 3c). Similarly, RC KI mice failed to reach the performance of their dopamine D1 and D2 receptor antagonist treated WT controls (trial × genotype × drug interaction: $F_{6,116} = 2.24$, $p = 0.044$; $latency_{WT\_antagonists}$ vs $latency_{RC\_antagonists}$, $p = 0.043$, 3-way ANOVA with Tukey *post hoc* test) (Fig. 3c). Overall, the data here indicate that dopamine receptor antagonism during the skill acquisition impeded the initial motor performance even after dopamine signaling restoration. However, the performance of WT and GS mice gradually improved over time; on the contrary, the latency of fall in the RC mice remained degraded.

To rule out the possibility that the impaired learning of RC mice was due to delayed clearance of the antagonists, a subgroup of RC mice that received the administration of dopamine receptor antagonists were returned to their home cages without rotarod training (here referred to as "untrained mice"). After the 72-h break, these mice were tested on the rotarod task; their performance was improved compared to the RC mice that underwent training in the rotarod task immediately following systemic administration of dopamine receptor antagonists, as measured by latency to fall (main effect of treatment, $F_{1, 260} = 189.2$, $p < 0.0001$, two-way ANOVA) (Fig. 3d). These findings suggest that the impeded motor learning in the RC mice resulted from the dopamine antagonism and training combination. To examine the contributions of the dopamine receptor subtypes in shaping motor learning on the rotarod task, we administered D1 and D2 receptor antagonists separately during the initial acquisition. The administration of SCH23390 or eticlopride impaired the performance during the initial phase of the rotarod training across all genotypes (session 1–5) compared to their saline-treated controls (Supplementary Fig. 6a, c). Mice previously treated with D1 receptor antagonist showed immediate rotarod improvement after 72 hr washout, regardless of genotype (Supplementary Fig. 6a, b, Supplementary Table 7), suggesting a limited contribution to the aberrant learning. However, D2 receptor antagonism resulted in a delayed improvement in performance (Supplementary Fig. 6c, d, Supplementary Table 7). While we found no change in the performance in the averaged last three sessions across genotypes (Supplementary Table 7), WT and GS mice treated with D2 receptor antagonist reached the performance of their saline controls at the last session of the task (session 18: $latency_{GS\_eticlopride} = 111.9 \pm 9.1$ s, $latency_{GS\_saline} = 139.1 \pm 10.5$ s; $p = 0.5789$, $latency_{WT\_eticlopride} = 135.2 \pm 23.1$ s, $latency_{WT\_saline} = 133.8 \pm 15.2$ s; $p = 0.9999$, 2-way ANOVA, Sidak's *post hoc* test). In contrast, RC mice did not exhibit the performance of their saline controls for the last studied session ($latency_{RC\_eticlopride} = 106.3 \pm 14.3$ s, $latency_{RC\_saline} = 173.3 \pm 15.2$ s $p = 0.0042$; 2-way ANOVA with Sidak's *post hoc* tests). Given that LRRK2 mutations stimulate LRRK2 kinase activity, we examined whether increased kinase activity in RC mice contributes to this motor learning deficit. Therefore, we treated RC mice with the LRRK2 inhibitor MLi-2[55] prior to D1 and D2 receptor antagonism throughout the task. We found motor improvement across behavioral sessions during the drug-free phase in MLi-2-treated mice; this was evident with increased latency to fall across successive trials (time × treatment interaction, $F_{17,340} = 4.951$, $p < 0.0001$, two-way ANOVA) (Supplementary Fig. 6e). This inference was confirmed by analyzing the average latency in mice with and without MLi-2 treatment.

Increase in latency was only observed following MLi-2 injections (time x treatment interaction, $F_{3, 60} = 10.98$, $p < 0.001$; for MLi-2 with D1 + D2 antagonism treated RC mice $latency_{session\_7-9} = 38.7 \pm 7.0$ s vs $latency_{session\_10-12} = 67.7 \pm 6.4$ s; $p = 0.0002$, vs $latency_{session\_13-15} = 90.6 \pm 7.9$ s; $p < 0.001$, vs $latency_{session\_16-18} = 112.8 \pm 11.0$ s; $p < 0.001$, two-way ANOVA with Sidak's *post hoc* tests) (Supplementary Fig. 6d). The pre-treatment of MLi-2 decreased LRRK2 S935 phosphorylation—a well-characterized readout of LRRK2 kinase activity[56,57]—in striatal extracts (Supplementary Fig. 6e). Taken together, the accelerated rotarod paradigm with dopamine antagonist treatment unmasked kinase-mediated deficits of striatal motor learning in RC mice.

To show that locomotor deficits did not confound the motor learning impairment in RC mice, we examined the locomotor behavior of mice in the open-field arena. A subset of WT, RC, and GS mice that underwent dopamine antagonism during the initial 5-day rotarod training were assessed by the total distance traveled following the 72 h break of the rotarod task. We found no differences between any of the groups (Fig. 3e). In contrast, acute D1 and D2 receptor antagonism, 2 h before open field, substantially reduced open-field activity (inset Fig. 3e). In summary, disruption of dopamine signaling during the rotarod training task specifically interferes with striatal motor learning in the RC mice, which otherwise exhibit no deficits in naturalistic behavior such as locomotion even after restoring dopamine signaling.

**Increased synaptic PKA activities in the striatum of R1441C mice following striatal motor learning**. To identify the signaling pathways that underlie the motor learning deficits, specifically in the RC mice we performed six-plex tandem mass tag (TMT) quantitative mass spectrometry (MS) to compare the protein expression of either RC or GS and WT mice following the first five days of the rotarod test. P2 crude synaptosomal preparations (Supplementary Fig. 7a) of all genotypes were prepared. Each of the WT, RC, and GS groups were labeled with TMT and then combined and analyzed by MS. Differentially expressed proteins were quantified by comparing the normalized average reporter ion intensities of peptides among the three biological replicates from pairwise comparisons of WT and RC as well as WT and GS striatal synaptosomal fractions (Fig. 4a). In total, 972 proteins were identified. Proteins of relative quantitation were divided into two categories. A quantitative ratio over 1.2 was considered upregulation, while a quantitative ratio less than 1/1.2 was considered as downregulation ($p < 0.05$) (Fig. 4b, c). The number of differentially expressed proteins is summarized in Fig. 4b, c. We annotated their features using the KEGG database (Fig. 4d, Supplementary Fig. 8). Top enriched pathways (strength ≥ 1) and network analysis for differentially expressed proteins between RC, WT, and GS synaptosomal fractions are shown in Supplementary Fig. 8. A complete list of altered proteins and pathways can be found in Supplementary Datas 1 and 2. One of the top enriched pathways in RC mice was the cAMP-PKA signaling pathway, a master signaling pathway in the SPNs[58]. Our previous findings show that synaptic PKA signaling was increased in the RC and not GS striatal extracts[26]. We hypothesized that the PKA signaling pathway might explain the specific RC effect on motor learning. To this end, we used a pan phospho-PKA substrate antibody to detect phosphorylation of downstream PKA targets in the synaptosomal preparations of WT, RC, and GS mice trained in the rotarod task and ultimately validate the MS data. We confirmed an increase in the PKA activities in the RC mice and not GS mice compared to control (one way-ANOVA, $p = 0.015$; *post hoc* tests as noted, $n = 5$ mice for each group) (Fig. 4e, f). The phospho-PKA levels in the striatal synaptosomal

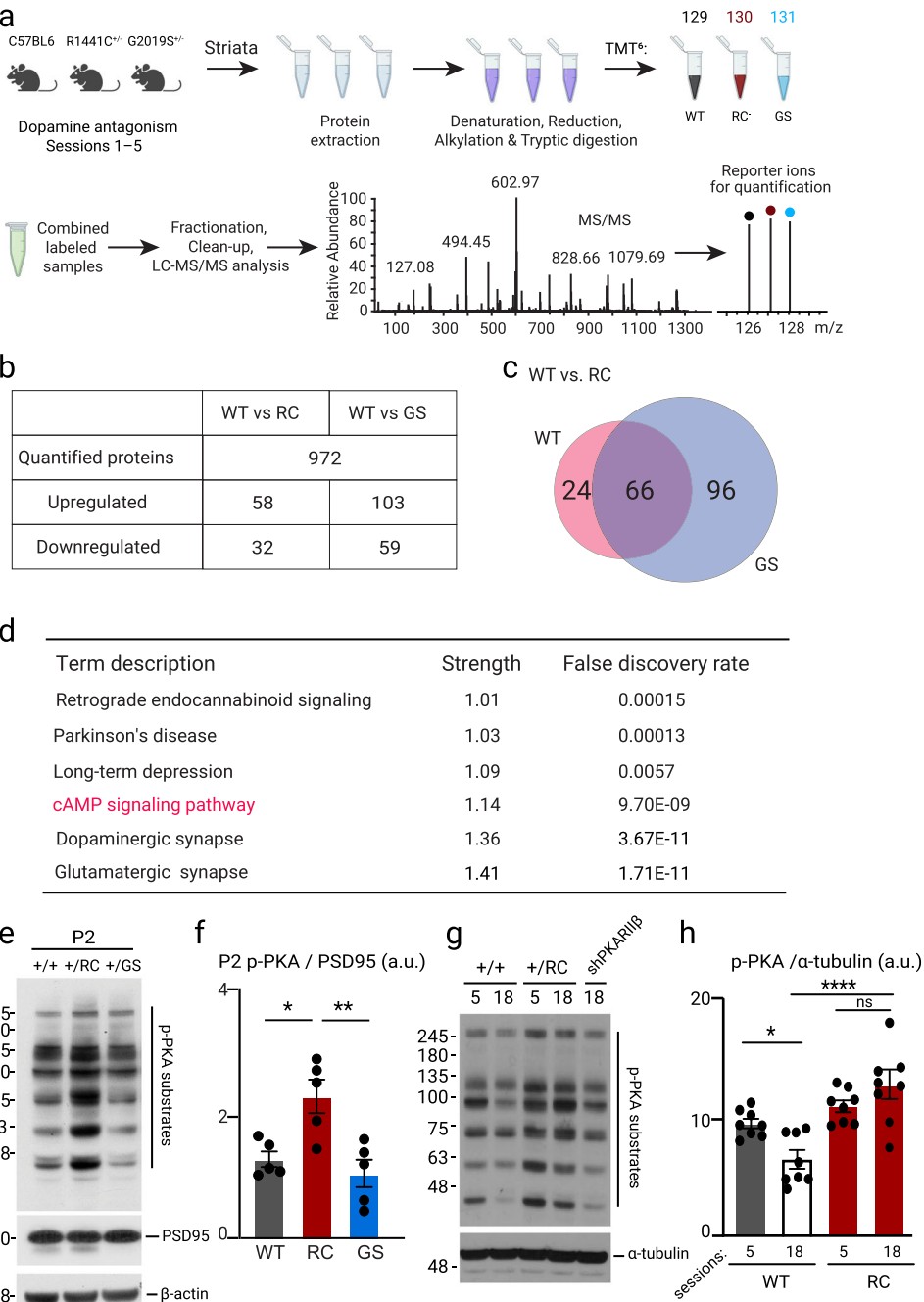

**Fig. 4 Synaptic PKA signaling is increased in the striatum of RC mice trained in the rotarod task. a** Workflow of proteomic screening for total TMT-labeled peptides in WT, RC, and GS striatal crude synaptosomal fraction after 5 days of cocktail (D1R and D2R antagonists) administration and rotarod training. **b** Pairwise comparisons of WT vs. RC and WT vs. GS proteins revealed that 90 proteins for WT vs. RC and 162 proteins for WT vs. GS were differentially regulated. These findings were further quantified if the proteins were up- or down-regulated. **c** Venn diagrams show the overlap of modulated (both up and down-regulated) proteins between WT versus RC and WT versus GS comparisons. 66 differentially regulated proteins were common between the WT vs. RC and WT vs. GS groups. **d** KEGG analysis of selective pathways altered in RC synaptosomal fraction compared to WT. The top enriched pathways are found in Supplementary Fig. 8, while the full lists of differentially expressed proteins and pathways in WT versus RC and WT versus GS can be found in Supplementary F 1 and 2. **e** WB analysis of striatal P2 fraction of WT, RC, and GS mice after the rotarod paradigm probed for p-PKA, PSD95, and β-actin. **f** Quantification of p-PKA band intensities normalized to PSD95. Summary graphs represent the mean, while error bars SEM, $*p < 0.05$, Tukey post hoc following one-way ANOVA ($n = 5$). **g** Western blot analysis of striatal extracts of D1 + D2 receptors antagonists administered WT, RC and RC injected with shPKARIIβ probed for p-PKA substrates. Session 5 refers to the end of the drug phase of the rotarod, whereas session 18 denotes the end of the rotarod behavioral test. **h** Quantification of p-PKA bands normalized to α-tubulin in striatal extracts of WT and RC mice after session 5 and 18 of the rotarod task described in Fig. 3. Summary graphs reflect the mean, and error bars reflect SEM. $*p < 0.05$, $****p < 0.0001$, Tukey post-hoc test following one-way ANOVA ($n = 3$).

preparations were decreased at the end of the task in the WT mice ($WT_{session5} = 9.47 \pm 0.39$ a.u., $WT_{session18} = 6.07 \pm 0.71$ a.u.), $p = 0.02$, 1-way ANOVA, Tukey post-hoc test (Fig. 4g, h) at a time point, their initial impaired performance was improved, similar to their saline controls. However, the PKA signaling remained elevated in the RC mice throughout the behavioral task ($RC_{session5} = 10.93 \pm 0.498$ a.u., $RC_{session18} = 12.75 \pm 1.20$ a.u.) and higher than WT after session 18, ****$p < 0.0001$, 1-way ANOVA, Tukey post-hoc test (Fig. 4g, h) correlating with their impaired performance in the rotarod test (Fig. 3b, c). These findings suggest a strong correlation between phospho-PKA levels and striatal motor learning performance in the behaving mice.

**Aberrant PKA signaling underlies the motor deficits of R1441C mice.** To directly connect aberrant PKA activities and performance impairment of the RC mice in the behavioral test, we decided to determine if manipulation (i.e., decrease) of PKA signaling reverses the impairment of their performance in the rotarod test. PKA is a holoenzyme that consists of two regulatory and two catalytic subunits[59]. PKARIIβ is the dominant regulatory subunit in the striatum and genetic targeted disruption of the RIIβ gene in mice leads to a dramatic reduction in total PKA activity in the striatum[60]. Therefore, we reasoned that a decrease in the expression of the PKARIIβ protein in the striatum would result in a marked reduction in striatal PKA activity. To address this, we used viral-mediated RNA interference to knock down the gene encoding the PKARIIβ within the striatum of RC mice (see Methods). The successful administration of the AAVs was evidenced by eGFP fluorescence in the striatal sections of injected mice (Fig. 5a). Western blot analysis confirmed a 45% decrease in PKARIIβ protein levels in the shPKARIIβ injected mice compared to control injected ones, $p = 0.0015$, unpaired t-test (Fig. 5b, c). The reduced levels of the pan phospho-PKA antibody signal and PKA-mediated phosphorylation of S845-GluA1 in the striatal extracts of the sh-PKARIIβ injected mice indicate decreased PKA signaling in the striatum of the mice after the PKARIIβ subunit knockdown (Fig. 5d).

While the sh-PKARIIβ similarly decreased the PKARIIβ protein across WT and RC mice (Fig. 5e, f), it returned the aberrant p-PKA signaling in RC mice back to WT's sh-control level ($p$-$PKA_{RCsh-PKARIIβ} = 1.1 \pm 0.04$ a.u, $p$-$PKA_{WTsh-control} = 1.09 \pm 0.08$ a.u) (Fig. 6a, b). Notably, the sh-PKARIIβ viral construct administration in the WT mice ($p$-$PKA_{WTsh-PKARIIβ} = 0.43 \pm 0.05$ a.u) reduced the PKA signaling to a lower level compared to both WT sh-control ($p$-$PKA_{WTsh-control} = 1.09 \pm 0.08$ a.u, $p = 0.0004$) and sh-PKARIIβ injected RC mice ($p$-$PKA_{RCsh-PKARIIβ} = 1.1 \pm 0.04$, $p = 0.0003$, one-way ANOVA, Tukey post-hoc test (Fig. 6a, b).

sh-PKARIIβ reversed the impaired performance of the RC mice in the presence of the D1 and D2 antagonists. Specifically, we observed an impaired performance in the sh-control injected RC mice treated with the antagonists cocktail, consistent with the decreased latency to fall of the uninjected RC mice previously described in Fig. 3b, c (sessions 16–18; $latency_{RC\_saline} = 97.0 \pm 5.7$ s, $latency_{RC\_antagobnists\_shcontrol} = 41.8 \pm 5.0$ s, $p < 0.0001$, one-way ANOVA, Fig. 6c, e). In contrast, the performance of the mice injected with the sh-PKARIIβ virus was improved and found to be similar to the saline control RC mice (sessions 16–18; $latency_{RC\_saline} = 97.0 \pm 5.7$ s, $latency_{RC\_antagobnists\_shPKARIIβ} = 76.5 \pm 7.2$ s, Fig. 6c, d). The centrality of the PKA signaling in the specific dopamine-dependent striatal motor learning task was confirmed by the impaired reduction of a subset of antagonists treated and trained WT mice administered the sh-PKARIIβ viral constructs sessions compared to antagonists treated ones (sessions 16–18; $latency_{WT\_antagonists} = 126 \pm 11.6$ s, $latency_{WT\_antagobnists\_shPKARIIβ} =$

$54.1 \pm 9.9$ s, $p < 0.0001$, one-way ANOVA) (Fig. 6d, f). Our findings argue that balanced synaptic PKA signaling is critical for the dopamine-dependent striatal motor learning task.

## Discussion

The present study systematically investigated the molecular, physiological, and behavioral alterations in the striatum mediated by two different LRRK2 pathogenic mutations in KI mouse models. While a number of transgenic animal models have been generated to interrogate dysfunction associated with mutant LRRK2, the findings so far have been inconsistent across studies. The most parsimonious explanation is the difference in expression levels of mutant LRRK2 in the presence of endogenous LRRK2 and the different promoters used to drive mutant protein expression[17]. In this study, by using RC and GS KI mice that express the mutant LRRK2 protein with endogenous expression patterns and levels, we attempted to resolve previous conflicting reports of dopamine transmission[20,42]. We focused on heterozygous LRRK2 KI mice as homozygous and heterozygous LRRK2 mutation carriers exhibit similar clinical manifestations[61], and the GS and RC LRRK2 mutations are inherited as a Mendelian dominant condition[6,62]. While we found a decrease in nigrostriatal dopamine release in both RC and GS mice, we observed a decrease in excitability selectively in the iSPNs and not dSPNs of the RC mice. No differences were found in either dSPNs or iSPNs of the GS mice. The alterations in the cellular properties of iSPNs in RC mice were paralleled with their impairments in dopamine-dependent striatal motor learning. The observed changes demonstrate that LRRK2 mutations similarly impact the pre-synaptic dopamine release; however, they exhibit different modes of dysfunction postsynaptically, with RC mutations showing stronger alterations. This is consistent with our previous findings that LRRK2 shaped corticostriatal synaptic function in a mutation-specific manner[26].

Taken together, our analysis comparing the RC and GS KI mouse lines side by side shows that both LRRK2 mutations cause deficits in evoked dopamine release deficits. Previous studies that measured dopamine in GS KI mice have contradicted one another. Using microdialysis, reduced extracellular levels of dopamine at twelve months but not six months were reported[20]. In contrast, Tozzi et al. reported reduced striatal dopamine levels in mice at six months[42]. Moreover, a recent report showed no difference in peak dopamine release in slices of three months[25]. There are several possibilities for the variability across the results, including the age of animals, genetic backgrounds, and assessment methods to evaluate dopamine content and heterogeneity across striatal subregions of dopamine release in different experiments. We also showed a decrease in evoked dopamine release in the RC KI mice, similar to that observed in the GS mice. Our results are at odds with an earlier report that measured basal dopamine content in RC KI mice and found no changes using bulk tissue HPLC, which lacks the spatial and temporal resolution that fast-scanning cyclic voltammetry offers[19].

On the other hand, our data corroborate the finding that stimulated catecholamine release in cultured chromaffin cells of RC mice had a 50% reduction in dopamine release[19]. The decrease in evoked dopamine release could be attributed to the role of LRRK2 in regulating the presynaptic vesicle cycle[64]. Both RC and GS mutations decreased synaptic vesicle endocytosis[65]. Although the mechanism of the endocytosis dysregulation in the LRRK2 context is unclear, it has been suggested that the aberrant phosphorylation of Rab5b[65], endophilinA[66], auxillin[67], and synaptojanin-1[68,69] by the LRRK2 mutations may mediate these effects. While the impact of LRRK2 in the vesicular trafficking has been shown in different neuronal types, a recent report showed

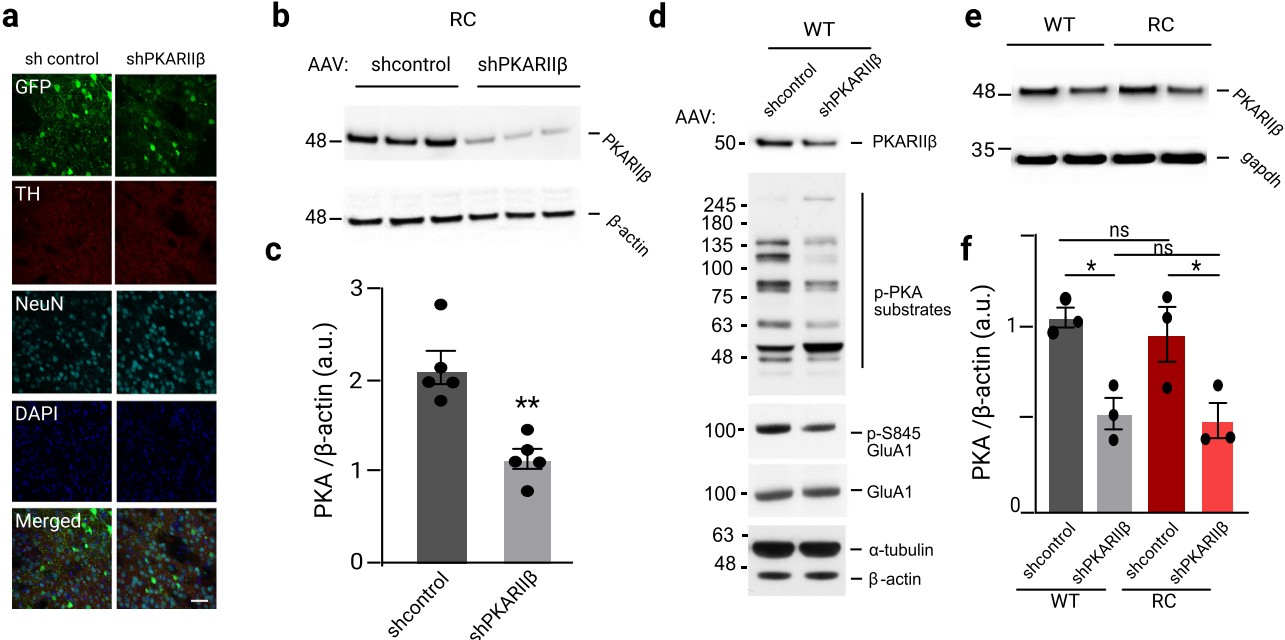

**Fig. 5 PKARIIβ knockdown modifies p-PKA signaling in the striatum of WT and RC mice. a** Striatal sections of AAV-eGFP-shcontrol and AAV-eGFP-shPKARIIβ injected WT and RC mice. Scale bar, 50 µm. **b** Western blot analysis of striatum of WT mice injected with AAV-sh-control and AAV-sh-PKARIIβ viral constructs probed for PKARIIβ. **c** Quantification of PKARIIβ from b normalized to β-actin. Summary graphs represent the mean, while error bars SEM, $**p < 0.01$, unpaired t-test ($n = 5$). **d** Western Blot analysis of striatum of WT mice injected with, AAV-sh-control and AAV-sh-PKARIIβ probed for PKARIIβ, pPKARIIβ, p-S845GluA1, GluA1 and β-actin. **e** Western blot analysis of striatal extracts from WT and RC mice injected with AAV-sh-control and AAV-sh-PKARIIβ viral constructs probed for PKARIIβ, and gapdh. **f** Quantification of PKA bands normalized to gapdh. Summary graphs represent the mean, while error bars SEM, $*p < 0.05$, $**p < 0.01$, Tukey post hoc following one-way ANOVA ($n = 3$).

slower exocytosis, which could be reversed by LRRK2 kinase inhibitors, only in primary dopaminergic and not cortical or hippocampal neurons of transgenic GS mice[69]. It is intriguing to hypothesize that presynaptic vesicle cycle dysregulation may contribute to the vulnerability of dopamine neurons in PD.

Given the importance of nigrostriatal dopamine signaling in striatal motor learning, we assessed motor learning in RC and GS KI lines using the accelerated rotarod. Consistent with previous studies, we observed no abnormalities in motor learning in LRRK2 mutants under basal conditions[19,54]. Dopamine receptor antagonism caused both a direct performance impairment and inhibitory learning that degraded motor performance even after restoring dopamine signaling on a rotarod motor-learning task[48,49]. This dopamine-dependent rotarod task unmasked alterations in the striatal motor learning specific to the RC mice. Unlike WT and GS mice, which gradually improved over sessions[48,70], RC mice failed to improve their performance with time. This learning impairment is experience-dependent, task-specific, and involves corticostriatal plasticity[48,49,71]. The different pattern of recovery–no effect of D1 antagonist vs. impaired performance after D2 antagonism–in the early drug-phase sessions suggest primarily a D2 receptor-mediated effect. In RC mice, there is little, if any, contribution of D1 receptor signaling (Supplementary Fig. 6). The performance of the RC D2 antagonist-treated mice appeared degraded compared to their corresponding saline controls; this change in performance only reached significance for the last testing session. Overall, the impaired performance of RC mice was stronger when both D1 and D2 receptors were antagonized, suggesting a synergistic effect between the two receptor subtypes.

We currently do not fully understand the dissociation between impaired dopamine release and striatal alterations. One explanation is that RC and GS mutations have distinct aberrant coupling to molecular signaling pathways in the SPNs. It is known, for instance, that RC and GS mutations affect substrate phosphorylation differently[27]. These distinct phosphoproteomes across mutations might be specific to SPN neurons where LRRK2 is highly expressed. Our earlier data support this idea that synaptic PKA signaling is selectively increased in RC and not GS striatal synaptic fractions[26]. We performed quantitative proteomics studies to gain insights into mutation-specific effects in the SPNs signaling landscape, particularly during the striatal motor learning task. Extending on our earlier data, we found that the synaptic PKA activity was elevated in the RC and not GS mice after the behavioral paradigm. PKA directs several critical striatal functions through its phosphorylation of target proteins[35,72]. For example, cAMP-PKA signaling is critical for striatal motor learning[60,73]. Overall, our findings suggest that too much or too little PKA signaling impairs motor performance. By suppressing PKA activity in the striatum of RC mice using viral mediated knockdown of a PKA subunit, we showed that increased postsynaptic PKA signaling in the RC mice underlies their impaired performance. Accordingly, earlier data from animal studies have shown that adenosine A2A receptor antagonists ameliorate the dopamine dependent inhibitory motor learning by decreasing cAMP signaling in the SPNs, emphasizing the centrality of the PKA pathway in these regulations[49,51]. This striatal targeted viral approach further supports that the motor learning deficits associated with the RC mutation is due to the striatum and not other brain regions, most notably the cerebellum, which plays a complementary role to striatal learning[74,75]. Given the opposing PKA signaling properties after dopamine signaling, future studies employing PKA sensors[63,76] are required to elucidate the SPNs subtype specific PKA signaling dysregulations in the RC mice. The application of the MLi-2 LRRK2 kinase inhibitor reversed the degraded performance in the RC mice, suggestive of kinase-dependent mechanisms. While several lines of evidence suggest a cross-

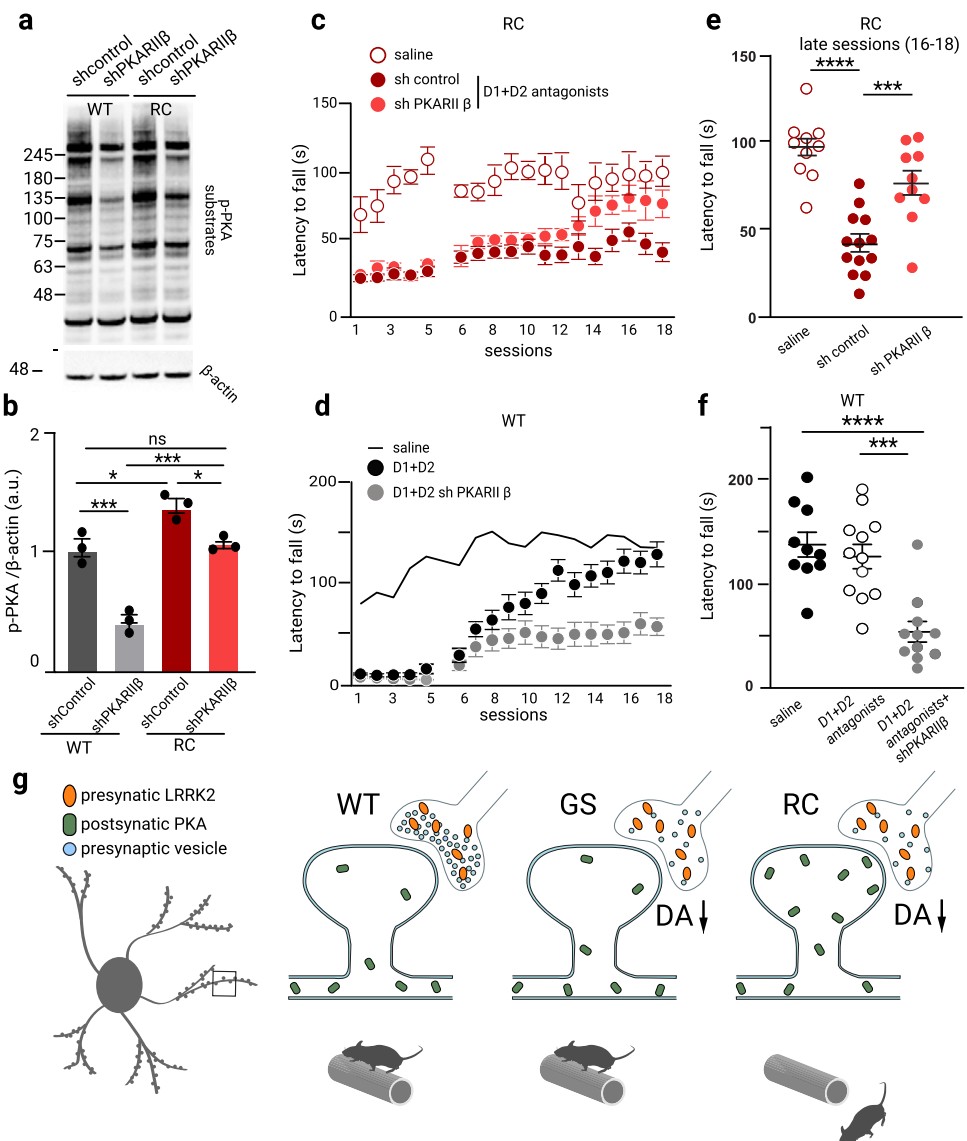

**Fig. 6 Aberrant synaptic PKA activities underlie the dopamine-dependent RC impaired motor learning. a** Western blot analysis of striatal extracts from WT and RC mice injected with AAV-sh-control and AAV-sh-PKARIIβ viral constructs probed for p-PKA and β-actin. **b** Quantification of p-PKA bands normalized to β-actin. Summary graphs represent the mean, while error bars SEM, *$p < 0.05$, ***$p < 0.001$, Tukey post hoc following one-way ANOVA ($n = 3$). **c, d** The latency to fall of RC mice treated with the dopamine receptors antagonists' cocktail and injected either with control AAV-sh-control and AAV-sh-PKARIIβ viruses was assessed in the rotarod paradigm described in Fig. 3a. A subset of mice was administered saline and used as a reference for the effect of D1 and D2 antagonism (saline treated group $n_{RCsaline} = 10$, D1 + D2 antagonists treated groups $n_{RCsh-control} = 13$, and $n_{RC-shPKARIIβ} = 10$). The average latency in the drug-free recovery phase of the three late sessions (16–18) are summarized in **e**. Asterisks show statistical significance for Tukey's multiple comparison tests after two-way ANOVA. **d** Dopamine receptor antagonists treated WT mice were injected with shPKARIIβ virus, and their latency to fall was assessed in the accelerating test. WT mice administered saline and cocktail are used as reference, saline treated group $n_{WT} = 10$, D1 + D2 receptor antagonist treated groups $n_{RC} = 12$, and $n_{RC-shPKARIIβ} = 11$. The average latency in the three late sessions (16–18) of the rotarod test is shown in **f**. Asterisks show statistical significance for Tukey's multiple comparison tests after two-way ANOVA. **g** Working model of aberrant PKA synaptic activities at a blowup of a spine of an SPN (left). Presynaptic dysregulation of LRRK2 activity in RC and GS mutants has a common disruption of the synaptic vesicle cycle, which leads to decreased dopamine release. In addition, in RC mutants, there is a postsynaptic translocation of PKA into the dendritic spines with aberrant increased PKA activity, which disrupts synaptic signaling and interferes with motor learning and performance (measured by decreased latency to fall off rotarod).

talk between LRRK2 and PKA[77,78], the precise mechanisms underlying the linkage of the two kinases are not simple and remain unknown. In addition, due to technical limitations, it remains to be shown whether iSPNS versus dSPNs in RC mice have differentially regulated PKA activity compared to WT mice. Such changes would affect motor performance, as a concurrent and coordinated balance between direct and indirect pathways is required for normal basal ganglia function[79,80]. A working model

of aberrant synaptic PKA signaling based on the results of our study is presented in Fig. 6g.

Our data from the KI mouse lines are of particular importance as they mirror subclinical dopaminergic dysfunction and corticostriatal alterations of the asymptomatic LRRK2 mutation carriers[81–83] in the absence of apparent neurodegeneration. A recent study showed that dopamine release deficits at the axonal terminals are paralleled with striatal motor learning deficits but

have no typical PD motor impairments in a PD model[84] in which a Parkinsonian phenotype emerges later. Thus, the investigation of these dopamine and striatal dysfunctions that characterize the prodromal PD in the mutant LRRK2 KI mouse provides the framework for implementing neuroprotective therapies and developing biomarkers to detect and monitor disease progression related to LRRK2 mutations. Given the striatal alterations observed in the RC mutant mice, our study highlights the importance of future studies focused on the mutations of the GTPase domain and its downstream signaling targets for the development of signaling-specific neuroprotective therapies. A clear understanding of the different impacts of LRRK2 pathogenic mutations will enable patients' stratification and personalized therapeutic strategies for PD manifesting LRRK2 mutation carriers.

## Method

**Mice**. All experiments were in compliance with Northwestern University Animal Care and Use Committee guidelines. For electrophysiological studies, Drd1a-tdTomato mice (Jackson Laboratory 016204, RRID:IMSR_JAX:016204) were crossed with R1441C mice (Jackson Laboratory 009346, RRID:IMSR_JAX:009346)[19], and Drd2-eGFP mice (MMRC 000230, RRID:MMRRC_000230-UNC) were crossed with G2019S mice RRID:IMSR_JAX:030961)[20]. All mice were maintained on the C57BL/6 (Jax 000664, RRID:IMSR_JAX:000664) background. Heterozygotes for mutant LRRK2 alleles and their littermate controls, or wild-type (WT) mice, were used in all experiments. For electrophysiological recordings, hemizygotes for Drd1a-tdTomato and Drd2-eGFP were used for cellular identification. Mice were group-housed on a standard 12/12 hr light/dark cycle. Both males and females were used in this study.

**Visualized ex vivo electrophysiology**. Mice at postnatal day 90–110 were anesthetized with a ketamine-xylazine mixture and perfused transcardially with ice-cold aCSF containing the following (in mM): 125 NaCl, 2.5 KCl, 1.25 NaH$_2$PO$_4$, 2.0 CaCl$_2$, 1.0 MgCl$_2$, 25 NaHCO$_3$, and 12.5 glucose, bubbled continuously with carbogen (95% O$_2$ and 5% CO$_2$). The brains were rapidly removed, glued to the stage of a vibrating microtome (Leica Instrument), and immersed in ice-cold aCSF. Parasagittal slices containing the dorsal striatum were cut at a thickness of 240 μm and transferred to a holding chamber where they were submerged in aCSF at 37 °C for 30 min and maintained at room temperature thereafter. Slices were then transferred to a small-volume (~0.5 ml) Delrin recording chamber mounted on a fixed-stage, upright microscope (Olympus). Neurons were visualized using differential interference contrast optics (Olympus), illuminated at 735 nm (Thorlabs), and imaged with a 60× water-immersion objective (Olympus) and a CCD camera (QImaging). Genetically defined neurons were identified by somatic eGFP or tdTomato fluorescence examined under epifluorescence microscopy with a white (6,500 K) LED (Thorlabs) and appropriate filters (Semrock).

Recordings were made at room temperature (20–22 °C) with patch electrodes fabricated from capillary glass (Sutter Instrument) pulled on a Flaming-Brown puller (Sutter Instrument) and fire-polished with a microforge (Narishige) immediately before use. Pipette resistance was typically ~3–4 MΩ. For whole-cell current-clamp recordings, the internal solution consisted of the following (in mM): 135 KMeSO$_4$, 10 Na$_2$phosphocreatine, 5 KCl, 5 EGTA, 5 HEPES, 2 Mg$_2$ATP, 0.5 CaCl$_2$, and 0.5 Na$_3$GTP, with pH adjusted to 7.25–7.30 with KOH. The liquid junction potential for this internal solution was ~7 mV and was not corrected. For voltage-clamp recordings, neurons were clamped at −80 mV with an internal solution that contained the following (in mM): 125 CsMeSO$_3$, 10 Na$_2$-phosphocreatine, 5 HEPES, 5 tetraethylammonium chloride, 2 Mg$_2$ATP, 1 QX314-Cl, 0.5 Na$_3$GTP, 0.25 EGTA, and 0.2% (w/v) biocytin, with pH adjusted to 7.25–7.30 with CsOH. Stimulus generation and data acquisition were performed using an amplifier (Molecular Devices), a digitizer (Molecular Devices), and pClamp (Molecular Devices). For current-clamp recordings, the amplifier bridge circuit was adjusted to compensate for electrode resistance and was subsequently monitored. The signals were filtered at 1 kHz and digitized at 10 kHz. KMeSO$_4$ and Na$_2$-GTP were from ICN Biomedicals and Roche, respectively. All other reagents were obtained from Sigma-Aldrich.

To determine the excitability of SPNs, the frequency-current (F-I) relationship of each cell was examined with current-clamp recordings. A series of 500 ms current steps of n were applied beginning at −150 pA and incremented at 25 pA for each consecutive sweep. This protocol was applied until each recorded cell reached maximal firing and entered a depolarization block. Resting membrane potential was monitored for stability, and cells that varied 20% from mean baseline were excluded from the analysis. For current-clamp recordings, the amplifier bridge circuit was adjusted to compensate for electrode resistance and monitored during recording. Recordings with more than a 10% change were not included as data used in the analysis.

Corticostriatal responses were recorded in voltage-clamp as previously described[39]. Electrical stimulation was performed using parallel bipolar tungsten electrodes (FHC) placed in layer 5 of the cortex. Stimulus width and intensity were adjusted via a constant current stimulator (Digitimer) to evoke a first excitatory postsynaptic current (EPSC) with an amplitude of 200–400 pA in the presence of the GABA$_A$ receptor antagonist SR95531 (10 μM) and CGP55845 (1 μM). Whole-cell access was monitored with a −5 mV pulse throughout the recording. Membrane capacitance (Cm) was determined off-line as Cm = Q$_t$ * V$_{test}$, where Qt was calculated as the integral of the transient current elicited by V$_{test}$, a 10-mV voltage step[85]. The paired-pulse ratio (PPR) for a given cell was calculated by taking the average of the ratios of the second EPSC amplitude to the first EPSC amplitude for each recording sweep. Data were excluded if the series resistance of the patch pipette differed by >20% between the two recordings. This protocol has been deposited to protocols.io with https://doi.org/10.17504/protocols.io.81wgby7zovpk/v1.

**Fast-scanning cyclic voltammetry**. Brain slices were prepared as described above in the electrophysiology section. Carbon fiber (7 μm diameter) (Goodfellow) electrodes were fabricated with glass capillary (Sutter) using a puller (Narishige), and fiber tips were hand-cut to 30–100 μm past the capillary tip. The carbon-fiber electrode was held at −0.4 V before each scan. A voltage ramp to and from 1.2 V (400 V/s) was delivered every 100 ms (10 Hz). Before recording, electrodes were conditioned by running the ramp at 60 Hz for 15 min and at 10 Hz for another 15 min and calibrated using 1 μM dopamine hydrochloride (Sigma). Dopamine transients were evoked by electrical stimulation delivered through a concentric, bipolar electrode (FHC) placed in the rostrodorsal striatum (Fig. 1b) because of its known and important involvement in reward and motivation[86–89] and motor learning and control[89–91]. A single electrical pulse (300 μA, 0.2 ms) was used[92,93]. Data were acquired with an amplifier (Molecular Devices), a digitizer (Molecular Devices), and pClamp (Molecular Devices). For each slice, four measurements were made and then averaged. The custom analysis was written in MATLAB (Math-Works). The voltammogram and peak oxidative current amplitudes of the dopamine transient were measured. Experiments were rejected when the evoked current did not have the characteristic electrochemical signature of dopamine.

All voltammetric measures were performed by sampling at dorsostriatal sites ~200–400 μm ventral and posterior from the forceps minor corpus callosum. Four recordings were taken at each site with two-minute intervals between recordings and then averaged as a reported measure. To minimize potential confounds of regional differences in dopamine release[94], a systematic sampling of dopamine release across the lateromedial axis of the rostrodorsal striatum was performed in parasagittal slices. As we found no significant sex differences within WT and mutant mice, measures from both sexes were combined. The most lateral slice where the GPe was first evidenced was taken as the lateral slice. Consecutive slices 480 μm apart and medial of this lateral slice were taken as being intermediate and medial. Additionally, the recording order of the slices were randomized so as to reduce potential recording biases and mitigate possible electrode sensitivity issues. Electrode sensitivity was retested at the end of recordings with a freshly made dopamine stock used for calibration. Recordings from electrodes that had larger than 10% change were discarded. This protocol has been deposited to protocols.io with https://doi.org/10.17504/protocols.io.kxygx9py4g8j/v1.

**Behavioral tests**. Motor learning was assessed with an accelerating rotarod in WT, RC, and GS mice. The task started when the mice were around postnatal day 60 using a rotarod apparatus (Panlab) equipped with a mouse rod (3 cm diameter) and set to 4–40 rpm acceleration over 300 s. The task consisted of eighteen daily sessions (five trials per session; intertrial-interval = 15 s, max trial duratio$n$ = 300 s) divided into two phases[48] (Fig. 3a). Specifically, during the dopamine receptor antagonism phase (session 1–5), mice were systemically injected (i.p.) 30 min prior to testing with one of the following (prepared in 0.9% saline): cocktail of 1 mg/kg SCH23390 + 1 mg/kg eticlopride, 1 mg/kg SCH23390, or 1 mg/kg eticlopride. Following a 72-hr break, mice were then tested for another thirteen sessions (drug-free recovery phase). To determine if LRRK2 mutation contributes to motor learning deficit, a cohort of RC mice was administered with 5 mg/kg MLi-2 (LRRK2 inhibitor, Tocris Biosciences) 60 min prior to all of the 18 daily sessions. The inhibitor was dissolved in DMSO (25 mg/ml), and further diluted in 0.9% saline, followed by ultrasound[55,95]. The saline or cocktail of 1 mg/kg SCH23390 + 1 mg/kg eticlopride was given 30 min before the first five daily sessions. All the injections were given i.p. (0.005 ml/kg). A separate cohort of WT, RC, and GS mice underwent the accelerating rotarod task for the initial drug-treated phase. Before initiating the drug-free recovery phase (after the 72-hr break), mice were assessed individually in 56 ×56 cm open-field arenas in noise-canceling boxes and illuminated by dim red lights. The session (five-minute duration) started when mice were placed in the center of the arena. Locomotor activity was analyzed by the LimeLight 5 (Actimetrics, RRID: SCR_014254) software and reported as distance traveled. This protocol has been deposited to protocols.io with https://doi.org/10.17504/protocols.io.261ge345jl47/v1.

**Subcellular fractionation and Western blot analysis**. Subcellular fractionation of the mouse striatum was performed as previously described[23,26] (Supplementary

Fig. 7). We euthanized the mice and collected tissue 24 h after the 5th session. The tissue was kept frozen at −80 °C. Briefly, mouse striata were dissected and rapidly homogenized in four volumes of ice-cold Buffer A (0.32 M sucrose, 5 mM HEPES, pH7.4, 1 mM MgCl₂, 0.5 mM CaCl₂) supplemented with Halt protease and phosphatase inhibitor cocktail (Thermo Fisher Scientific) using a Teflon homogenizer (12 strokes). Homogenized brain extract was centrifuged at $1400\,g$ for 10 min. The supernatant (S1) was saved, and pellet (P1) was homogenized in buffer A with a Teflon homogenizer (five strokes). After centrifugation at $700\,g$ for 10 min, the supernatant (S1′) was pooled with S1. Pooled S1 and S1′ were centrifuged at $13{,}800\,g$ for 10 min to the crude synaptosomal pellet (P2) 20–40 μg of the supernatant were separated by 4–12% NuPage Bis-Tris PAGE (Thermo Fisher Scientific) and transferred to membranes using the iBlot nitrocellulose membrane Blotting system (Thermo Fisher Scientific) by following manufacturer's protocol. Primary antibodies specific for pS935 LRRK2 (Abcam Cat# ab230261, RRID:AB_2811274, 1:1000), total LRRK2 (Abcam Cat# ab133474, RRID:AB_2713963, 1:1000), and β-actin (Sigma-Aldrich Cat# A1978, RRID:AB_476692, 1:3000), Scientific MA1-045, 1:3000), dopamine D1R (Sigma-Aldrich Cat# D187, RRID:AB_1840789), dopamine D2R (Frontier Institute Cat# D2R-Rb, RRID:AB_2571596), DAT (Millipore Cat# MAB368, RRID:AB_94947), p-T74DARPP-32 (Cell Signaling Technology Cat# 12438, RRID:AB_2797914), total DARPP-32 (Cell Signaling Technology Cat# 2306, RRID:AB_823479), PSD95 (Thermo Fisher Scientific Cat# MA1-045, RRID:AB_325399, 1:1000), p-PKA substrates (Cell Signaling Technology Cat# 9624, RRID:AB_331817) were used. Secondary anti-mouse (Thermo Fisher Scientific Cat# PA5-62650, RRID:AB_2649666) and anti-rabbit antibodies (Thermo Fisher Scientific Cat# 65-6120, RRID:AB_2533967) were from Invitrogen. Membranes were incubated with Immobilon ECL Ultra Western HRP Substrate (Millipore) for 3 min prior to image acquisition. Chemiluminescent blots were imaged with iBright CL1000 imaging system (Thermo Fisher Scientific).

**Viral-mediated short-hairpin RNA knockdown**. Plasmid adeno-associated viruses (pAAVs) for knocking down the mouse PKARIIβ gene were custom generated by Vector Biolabs. Six short-hairpin RNAs (shRNAs) were designed against mPRKAR2B (NM_011158), and each shRNA plasmid was co-transfected into HEK-293 cells (DSMZ Cat# ACC-305, RRID:CVCL_0045) with mPRKAR2B cDNA plasmid for comparing the knockdown efficiency. The knockdown efficiency of the mPRKAR2B-targeting shRNAs and a control shRNA (CAA-GATGAAGAGCACCAA) was measured using quantitative PCRs. The shRNA with the targeting sequence GGAAGATGTACGAGAGCTTTA showed the highest knockdown efficiency and was cloned into the AAV-GFP-U6 vector for AAV1 packaging (Cat no, 7040, Vector Biolabs). Unilateral stereotactic (David Kopf Instruments) injections of 500–750 nl of AAV1-GFP-U6-shPKARIIβ or AAV1-GFP-U6-sh control viruses (5 × 10^12 genome copies/ml) were performed in four sites in the striatum (anterior-posterior 0.8 mm mediolateral 2.4 mm, and dorsoventral −2.8 and −3.6 mm relative to bregma and anterior-posterior 0.2 mm mediolateral 1.8 mm, and dorsoventral −2.8 and −3.6 mm relative to bregma). The mice were injected with the viral constructs four weeks before the rotarod test. This protocol has been deposited to protocols.io with (https://doi.org/10.17504/protocols.io.q26g7y819gwz/v1).

**Quantitative proteomics and analysis**. Mouse striata of mice collected 24 h after the 5th session of the rotarod test were fractionated to obtain crude synaptosomes (P2) as described in the subcellular fractions and Western Blot section. P2 pellets were resuspended with binding buffer (50 mM Tris-HCl, pH 7.5, 1% triton-X-100, 150 mM NaCl, 1 mM EDTA, 1 mM AEBSF with protease inhibitor cocktail and phosphatase inhibitor) and solubilized for 1 h at 4 °C. The detailed information for Tandem Mass Tag-liquid chromatography/mass spectrometry (TMT-LC/MS) was described previously in detail[96]. Briefly, extracts isolated from tissues were reduced, alkylated and digested overnight. The samples were labeled with the tandem mass tag (TMT) sixplex Isobaric Label Reagent as follows: 129 for WT, 130 for RC, and 131 for GS for three biological replicates per the manufacturer's instructions (Thermo Scientific). They were then mixed before sample fractionation and clean-up. Specifically, each sample for each genotype (WT, RC, or GS) was isolated and mixed in a tube. This was repeated to establish three replicates per genotype. Peptides were analyzed by LC-MS/MS using a Dionex UltiMate 3000 Rapid Separation nanoLC and a Q Exactive HF Hybrid Quadrupole-Orbitrap Mass Spectrometer (Thermo Fisher Scientific Inc.). Proteins were identified from the tandem mass spectra extracted by Xcalibur version 4.0. MS/MS spectra were searched against the Uniprot Mouse database using Mascot search engine (Matrix Science; version 2.5.1). TMT reporter ion quantification and validation of identified peptides and proteins were performed by Scaffold software (version Scaffold_4.8.4, Proteome Software Inc.). The subject groups enabled two pairwise comparisons, for which the statistics of differential protein expressions were calculated using the LIMMA 3.46 (linear models for microarray data) package. For each comparison, a linear model was fitted to the expression data of the two considered subject groups only. Proteins with $|\log2(FC,\ \text{fold change})| > \log2(1.2)$ and $p < 0.05$ were considered as significantly differentially abundant. KEGG database was used to identify enriched pathways by a Fisher's exact test to test the enrichment of the differentially expressed protein against all identified proteins. The pathway with

a corrected $p < 0.05$ was considered significant. Except for enrichment analysis, these differentially expressed proteins are annotated in different pathways based on the definition of KEGG database. KEGG annotated pathways with strength (log10 observed/expected) ≥1 were considered the top enriched annotated pathways in the pairwise comparisons. Strength is a measure to describe how large the enrichment effect is. STRING database was used for known and predicted protein-protein interactions.

*Statistics and reproducibility*. General graphing and statistical analyses were performed with MATLAB2021b (MathWorks), SAS9.4 (SAS Institute), and Prism9 (GraphPad). Sample size (n value) is defined by the number of observations (i.e., neurons, cells, or mice) and is indicated in the legends. Unless noted otherwise, data are presented as median values ± median absolute deviations as measures of central tendency and statistical dispersion, respectively. Box plots are used for graphic representation of population data[97,98]. The central line represents the median, the box edges represent the interquartile ranges, and the whiskers represent 10–90th percentiles. Normal distributions of data were not assumed for electrophysiological data. Comparisons for unrelated samples were performed using Mann–Whitney $U$ test at a significance level (α) of 0.05. Unless noted otherwise, exact $P$ values are reported. Behavioral data and quantifications of WB are presented as mean ± standard error of the mean and were analyzed with either one-way two-way or three-way ANOVA with repeated measures followed by Tukey or Sidak's *post hoc* tests.

**Reporting summary**. Further information on research design is available in the Nature Research Reporting Summary linked to this article.

## Data availability
The mass spectrometry proteomics data have been deposited to the ProteomeXchange Consortium via the PRIDE partner repository with the dataset identifier PXD037003. Raw tabular data of electrophysiology, FSCV, and behavioral tests are available through the open access option on the Zenodo data repository site with the following (https://doi.org/10.5281/zenodo.7153552, and https://doi.org/10.5281/zenodo.7153453). The fast-scanning voltammetry, ex vivo physiology, behavioral tests, and the intracranial injections protocols have been deposited in protocols.io. DOIs are specified in the methods section. The uncropped WB blots are found in Supplementary Fig. 9. The lists of DEG proteins in Fig. 4 are in Supplementary Data 1, 2. The source data of Figs. 3,4,5, and 6 are in Supplementary Data 3, 4, 5, and 6 respectively.

## Code availability
Codes are openly available at Zenodo,:https://doi.org/10.5281/zenodo.7153493; https://doi.org/10.5281/zenodo.7153493; 10.5281.7153540.

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

## Acknowledgements

This work was supported by Michael J. Fox Foundation for Parkinson's Research (L.P.), NIH R01 NS097901 (L.P.), R01 NS097901 (C.S.C.), R01 NS069777 (C.S.C.), P50 NS047085 (C.S.C.), R01 MH109466 (C.S.C.), R01 NS088528 (C.S.C.), T32 NS041234 (H.S.X.), F32 NS098793 (H.S.X.). This research was funded in whole or in part by Aligning Science Across Parkinson's [ASAP-020600] through the Michael J. Fox Foundation for Parkinson's Research (MJFF). For the purpose of open access, the author has applied a CC BY public copyright license to all Author Accepted Manuscripts arising from this submission. We thank Brianna Berceau for colony management and technical support. We also thank Dr. Heather Melrose for providing the LRRK2 G2019S knock-in mice. We thank Brianna Berceau for colony management and technical support. We also thank Dr. Heather Melrose for providing the LRRK2 G2019S knock-in mice. Parts of Figs. 3, 4, and 6 were created with Biorender.com.

## Author contributions

H.S.X. and S.C. conducted the electrophysiological measurements. H.S.X. designed and conducted the voltammetric measurements. C.C., S.K., B.S., X.S., and G.S. conducted behavioral testing. G.L. performed immunofluorescence studies. H.S.X., C.C., and S.K. conducted the statistical analysis. L.P., C.S.C., and H.S.X. wrote the manuscript with input from all co-authors. L.P. and C.S.C. designed, directed, and supervised the project. All authors reviewed and edited the manuscript.

## Competing interests

The authors declare no competing interests.
