## [Peer Review File · Communications Biology]

Reviewers' comments:

Reviewer #1 (Remarks to the Author):

The authors investigated two LRRK2 mutations associated with PD, G2019S and R1441C on their effects on nigrostriatal function by examining dopamine release and SPN physiology as well as motor learning tasks in KI mouse models. They noted that both R1441C and G2019S mice showed comparably reduced dopamine release. On the other hand, the hypoexcitability of iSPN was noted only in RC mice. The reduction in excitability was noted by decreased evoked spikes along with increased membrane capacitance. The mice showed deficit in striatal dependent motor learning after a mixture of D1 and D2 antagonist treatment, which was rescued by Mli-2, a LRRK2 kinase inhibitor in RC mice.

These are interesting findings, supporting previous studies of LRRK2 R1441C effect on striatal physiology. However, the paper does not link the major findings of dopamine release deficit, iSPN hypoexcitability, and motor learning deficit in a mechanistic way. Is there a connection between dopamine release deficit and SPN physiological changes? Or do the authors speculate that the effect of mutations is on both presynaptic dopamine terminals and postsynaptic SPNs in a differential pattern?

Changes in DA (and adenosine) signalling through SPN GPCRs and PKA is thought to play a role in motor impairment from DA antagonist cocktail (Beeler et al. 2012). LRRK2-RC mutation, but not LRRK2-GS mutation, has been reported to reduce its interaction with PKARIIB by the authors (Parisiadou, 2014), which suggests a potential mechanism. However, the finding that MLI-2 could reverse motor impairment from DA antagonist seems to suggest that LRRK2's kinase activity may play a role instead. Could the authors comment on the link between LRRK2-RC mutation, LRRK2-PKARIIB interaction, and if it is known whether MLI-2 modulates LRRK2's interaction with PKARIIB?

In Fig. 2, other measures of intrinsic excitability such as resting membrane potential, input resistance, rheobase, etc. would strengthen the paper and perhaps shed light on underlying mechanism of LRRK2-RC mutant's reduced iSPN intrinsic excitability.

Control for nonspecific effect of Mli-2 would add strength to the conclusion about kinase activity of LRRK2 as a potential mechanism for motor learning deficit. Does the inhibitor restore the pLRRK2 levels in LRRK2-RC mutants back to WT levels? (Fig. 3d).

RC mice have deficient ability to recover motor learning after treatment with a mixture of D1 and D2 antagonists whereas GC has the same recovery as WT mice (Fig. 3). On the other hand, both RC and GS show deficit when treated with D2 antagonist (Fig. 3 Supplementary). In fact, the deficit is more apparent in GS mice. What is the interpretation on this discrepancy?

They note only N for recordings, but should note N for mouse # as well.

Fig. 3 Supp c, d showed early deficit in RC and GS, not late sessions. Inspection of 3c shows that differences more obvious in sessions 8 to 12 in-between early and late. What is the rationale for dividing them into these time periods? Is there overall time effect?

Technical points, typo etc.

- 1) Line 112, "mice was administered with 5 mg/kg MLI-2 (LRRK2 inhibitor)". What was MLI-2 dissolved in – saline or DMSO?
- 2) Line 113-114, "All the injections were given i.p (0.1 ml/kg)". This seems to translate to giving a 20 g mouse an IP injection of 0.002 ml = 2 ul, which seems quite low. Please double check.
- 3) Line 209 "SCHSCH23390": should be SCH23390?
- 4) Line 232-234, "Therefore, we treated RC mice with the LRRK2 inhibitor MLI-2 prior to D1 and D2 receptor antagonism. We found motor improvement across behavioral sessions during the drug-free phase of MLI-2-treated mice...". It is unclear through the paper if mice were treated with MLI-2 only for sessions 1-5 (as seems to be suggested by the above sentences), or if mice were treated with MLI-2 during sessions 1-18, as seems to be suggested by the method section that

MLi-2 was given "prior to all daily sessions" (line 112). The interpretation of LRRK2's role in rotarod learning vs. performance obviously depends critically on clearing this up.

5) Lines 260-261, "In contrast, RC mice did not exhibit the performance of their saline controls across sessions". However, the corresponding P value from Fig. 3 table supplement 1 (Session 16-18, RC, saline vs D2 antagonism) seems to be $P = 0.1492$?

6) Figure 3 table supplement 1: "D1 antagonism" column title was repeated twice. One of them probably should be "D2 antagonism".

Reviewer #2 (Remarks to the Author):

Regrets for the delay in returning the review.

Key findings in the manuscript by Xenias et al:

The purpose of the study is to identify electrophysiological differences in dopamine-related neurotransmission (and behavior) in *Lrrk2* KI mice as models of late-onset LRRK2-linked PD. The study is interesting and technically appears to be well done. Principal findings comprise:

- 1 - The paper demonstrates that p.R1441C and p.G2019S knock-in mutations in *Lrrk2* are responsible for a decrease in nigrostriatal dopamine release;
- 2 - Based on that finding the expectation that the authors had is that there would be differences in excitability of striatal projection neurons (SPN). To interrogate direct- and indirect-pathways, they tested the number of spikes in related projection neurons, i.e., dSPNs and iSPNs. The most puzzling finding of their work is that the number of spikes were lower only in p.R1441C but not p.G2019S mutant animals, and only in the iSPN category, but not in the dSPN-linked pathway. Nevertheless, there was a trend in p.G2019S mice;
- 3 - Increased membrane capacitance was found for iSPNs in p.R1441C mice, but not in p.G2019S mice;
- 4 - No difference in paired-pulse ratios (PRRs) but increased excitatory postsynaptic current (EPSC) decay time was observed in the iSPN pathway of p.R1441C mice;
- 5 - The authors suggest that *Lrrk2* kinase activity "we treated RC mice with the LRRK2 inhibitor MLI-2 [33] prior to D1 and D2 receptor antagonism" is the reason that underlies the differences seen vs. WT in the rotarod performance.

Comments and suggestions for edits / improvements:

- 1 - In their introduction, the authors did not mention other KI mice-based, similar investigations and related findings that had already been published by other authors, such as their (later) references #20 and #21;
- 2 - A more refined intro would also lend itself to a specific hypothesis statement for the proposed work, which was sadly missing;
- 3 - In their KI mice, have the authors convinced themselves that the two KI lines, i.e., GS and RC, indeed have the same mRNA and protein levels as do WT mice? This seems to be important given their different outcomes in electrophysiological tests. Herzig et al in 2011 have shown how distinct KI mutations can alter steady-states in different tissues;
- 4 - Was there a sex effect in fast scanning cyclic voltammetry? How many mice of each sex and what ages were used in their studies? The effect of sex in the context of some *Lrrk2*-linked phenotypes in mice has been published recently;
- 5 - Given the availability of antibodies now, it seems that measuring the levels of distinct p-Rab proteins in components of both the direct and indirect pathway from WT, RC and GS seems to be the logical next step to do in order to further elucidate the underlying mechanisms. Short of that a simple explanation for the interesting observation the authors made remains unanswered;
- 6 - In addition, could the authors provide evidence of the PKA pathway involvement in the p.R1441C driven outcomes as mentioned in the discussion, in order to further enrich the mechanistic insight?
- 7 - The experiments are done in heterozygous animals, which is laudable to mimic the human genotypic findings. The question arises: Does homozygosity of either RC or GS increase the measured outcomes (i.e., dopamine release, number of spikes, capacitance and rotarod performance)?

- 8 - Was the Mli-2 dissolved in saline or DMSO? If DMSO was used, figure 3 might benefit from having DMSO control. DMSO can change gene expression, see PMID: 30646511. Also, the legend for figure 3 should state the actual amount of administered Mli-2;
- 9 - At the Mli-2 dose used in the rotarod experiment, do the authors have knowledge of off target effects by Mli-2 ? From where was Mli-2 procured?
- 10 - The rationale for the usage of Mli-2 in p.R1441C mice treated with D1+D2 antagonist could be better explained;
- 11 - Other controls come to mind, for example, measuring dopamine release with and without Mli-2 (i.e., vehicle control), measuring the number of spikes in iSPNs with and without Mli-2, measuring capacitance of iSPNs with and without Mli-2.

Reviewer #3 (Remarks to the Author):

Xenias and co-authors investigate the effect of two distinct LRRK2 mutations, G2019S and R1441C, in dopaminergic neurotransmission. Using electrophysiological and behavioral analyses in knockin mice, they found that while both mutations cause dopaminergic deficits, they likely operate through different mechanisms. In particular, they observe that LRRK2-R1441C exhibits reduced nigrostriatal dopamine release in the indirect pathway which is accompanied by impaired motor learning. Administration of LRRK2 inhibitors rescues the behavioral deficits associated with DA-receptor antagonism in R1441C mice. Instead, despite the observed impairment in DA release in G2019S KI mice, no behavioral or electrophysiological alterations were found, at least in the analyses performed.

Although there are novel findings in this study related to the R1441C mutation, overall there is not substantial novelty and a comprehensive investigation of the molecular mechanisms underlying the observed phenotypes is missing.

I have a number of specific comments/suggestions to strengthen this study:

- Using fast scanning cyclic voltammetry, the authors found decreased evoked DA release in RC and GS slices, indicating that the DA signaling is impaired, as previously suggested at least for the G2019S mutation. At this point, to test at what level the impairment is, the authors should evaluate possible differences in DA signaling components, eg D1 and D2, pPKA/PKA, DARP32, pERK1/2/ERK1/2 etc It would be also interesting to measure DAT levels and activity.
- The AP recording graphs are unclear. There are no reference of the half-maximum firing in the graphs, eg in figure 2b the half-maximum firing rate does not correspond to 625 pA.
- To investigate whether the differences in excitability are due to differences in membrane size, they measured membrane capacitance and found increased C in the iSPNs of RC mice. Do these neurons present differences in arborization/spine number or maturation? Since they register in current clump, why not evaluating Na⁺ and Ca²⁺ currents, being those influencing AP? Why the capacitance of WT mice is different in supplemental figure 2A versus B? Are they using littermates?
- In addition of measuring evoked EPSCs, they should also investigate the spontaneous currents. Differences in spontaneous activity would support that the observed differences are post-synaptic.
- Behavior. Authors need to explain why they chose such an intense training (18 trials per day). How can they rule out that the observed differences at the rotarod are not due to differences in cerebellar activity?
- Mli2 treatment of R1441C mice before D1/D2 antagonism rescues the motor learning deficiencies. The rationale of using Mli2 is that R1441C has increased kinase activity. Can they show that the striata of these mice have more kinase activity (pS1292, pRab10)? As the G2019S are also hyperactive, what does Mli2 treatment do in the G2019S mice in the same type of test? And in WT?
- Using D1 and D2 antagonists separately the authors state that the effect depends on D2 but I don't see any statistical differences in the RC mice upon treatment (see supplemental figure 3 C and D). How do the authors interpret the delayed response in motor learning after administration of D1/D2 receptor antagonists in R1441C mice?
- In addition of using D1/D2 antagonists, DA receptor agonists should also be used. At low concentrations, pramipexole can fairly discriminate between D1 and D2.
- Unexpectedly there are no differences in the total distance travelled upon D1/D2 receptor

antagonists in any genotype (see supplemental figure 3f)? How can this be explained?
- Since the behavioural experiment suggests learning impairment in RC mice, it would be interesting to add a memory test (novel object recognition)

Reviewer #4 (Remarks to the Author):

In this study, Xenias et al examined dopamine transmission in the R1441C and G2019S LRRK2 KI mouse models. They found a decrease in nigrostriatal dopamine release in both lines. Furthermore, they detected a decrease in excitability of indirect-pathway striatal projection neurons in the R1441C mice, which was accompanied by impairments in dopamine-dependent striatal motor learning. Motor learning deficits were rescued by pharmacological LRRK2 kinase inhibition.

This work confirms previous data showing reduction of dopamine release in G2019S LRRK2 KI mice (PMID 25836420). Using R1441C KI mice used in this study, Tong et al were not able to detect significant changes in striatal dopamine levels (PMID: 19667187). However, Xenias et al employed fast-scanning cyclic voltammetry, which revealed significant decrease in DA level.

Overall, the findings are interesting. Importantly, instead of focusing on one mutation or the other, this study undertook a systematic comparison in two different mouse lines.

- It is unclear why only the R1441C mice develop LRRK2 kinase-dependent motor impairment.
- The authors speculate that the dissociation between impaired DA release and striatal alterations is linked to changes in striatal PKA signaling in the R1441C mice (line 286). It would be useful to discuss a bit more on the potential role of PKA in this context.
- The authors conclude "our data argue that the impact of LRRK2 mutations cannot be simply generalized". While it is already known that the impact of these two mutations is different, the authors should discuss more on the specific effects that they may have on different pathways and how this could provide opportunities for targeted therapies.

Response to Reviewers

We thank the Reviewers for their valuable feedback on our manuscript. It is important to us that we provide a clear and detailed presentation of our work. This feedback has helped strengthen our work by following the Reviewers' suggestions of examining the role of PKA. This new direction has given a mechanistic insight that unifies our studies as a whole. The work has now expanded with two additional main figures on the role of PKA and a behavioral rescue of learning. In addition, suggestions to expand our electrophysiological parameters were followed and included with additional supportive work as supplemental figures. The voltammetric dataset has also been expanded and examined for any sex-based differences (further discussed below in this response to the Reviewers). We have also streamlined the reporting of sample sizes by including now statistical summary tables for better readability. Accordingly, the manuscript has been rewritten and reorganized to reflect this expanded work and present a more cohesive development around PKA. We also include the limitations of our study in the Discussion section. Our point-by-point responses are given below.

Reviewer #1 (Remarks to the Author):

The authors investigated two LRRK2 mutations associated with PD, G2019S, and R1441C on their effects on nigrostriatal function by examining dopamine release and SPN physiology as well as motor learning tasks in KI mouse models. They noted that both R1441C and G2019S mice showed comparably reduced dopamine release. On the other hand, the hypoexcitability of iSPN was noted only in RC mice. The reduction in excitability was noted by decreased evoked spikes along with increased membrane capacitance. The mice showed deficit in striatal dependent motor learning after a mixture of D1 and D2 antagonist treatment, which was rescued by Mli-2, a LRRK2 kinase inhibitor in RC mice.

These are interesting findings, supporting previous studies of LRRK2 R1441C effect on striatal physiology. However, the paper does not link the major findings of dopamine release deficit, iSPN hypoexcitability, and motor learning deficit in a mechanistic way.

Is there a connection between dopamine release deficit and SPN physiological changes?

Response 1.1. Dopamine is known to modulate SPN physiology, including excitability^[1-3] and long-term depression^[4-6]. Related to the dopamine release deficits we report, reduced nigrostriatal dopamine is associated with reduction of iSPN excitability and increased corticostriatal synaptic input^[7-10]. While the reduction of dopamine in RC mice could lead to the observed decrease in excitability in iSPNs, the fact that no change in excitability or learning was observed in GS mice despite there also being dopamine release deficits suggests that changes unique to the RC mutant was causal to the alterations in membrane excitability and learning in RC mice.

We now show in this revised work that the RC mutation is associated with increased PKA activity at striatal synapses (**Figure 4**). This, in turn, negatively affects dopamine receptor signaling and synaptogenesis^[11]. In addition to increased activity of PKA reducing neuronal excitability^[12, 13], PKA is also a key regulator of dopamine-dependent learning^[7]. This argument is supported by the reversibility in the learning deficits of RC

mice by shRNA mediated knockdown of PKA (**Figure 5**). These new results are further elaborated in our revised and expanded Discussion.

Or do the authors speculate that the effect of mutations is on both presynaptic dopamine terminals and postsynaptic SPNs in a differential pattern?

Response 1.2. LRRK2 has been shown to regulate synaptic vesicle storage and recycling ^[14]. We speculate that aberrant phosphorylation of proteins at nigrostriatal terminals leads to the observed dopamine release defects in both RC and GS mice. This idea is supported by the finding that both RC and GS mutations result in decreased synaptic vesicle exocytosis ^[15].

While previous findings generally agree on LRRK2 mediated presynaptic vesicle trafficking dysregulation in nigrostriatal synapses across both RC and GS mutations, emerging evidence demonstrates mutation-specific dysregulation in the SPNs. For instance, our previous study showed that alterations in the SPNs synapses were stronger in the RC mice. Specifically, we observed enhanced single synapse glutamate uncaging-responses in RC SPNs, paralleled by an increased PKA dependent synaptic incorporation of the GluA1-containing AMPA receptors ^[16]. PKA signaling was specifically increased in the RC and not GS striatal extracts. Given the critical role of dopamine-dependent PKA signaling in the function of SPNs regulating an array of cellular functions ^[14], we hypothesize that increased PKA activity specific in RC iSPNs shapes their postsynaptic signaling differently compared to SPNs of the GS mutant. Such additional dysregulation of postsynaptic signaling in RC iSPNs would explain the motor learning deficits seen selectively in the RC mutation. We have included a working model in our revised manuscript (**Figure 5**).

Changes in DA (and adenosine) signaling through SPN GPCRs and PKA are thought to play a role in motor impairment from DA antagonist cocktail ^[17]. RC mutation, but not GS mutation, has been reported to reduce its interaction with PKARIIb by the authors ^[11], which suggests a potential mechanism. However, the finding that MLI-2 could reverse motor impairment from the DA antagonists would suggest that LRRK2 kinase activity may play a role instead.

Response 1.3. We agree with the Reviewer's comment that the differential effect of the LRRK2 mutations on PKA signaling underlies the motor impairment in the RC mice. We previously showed that LRRK2 regulates PKA signaling in the SPNs ^[11]. LRRK2, through its ROC domain (where the RC mutation is located), binds to the regulatory subunit of PKA (PKARII β) in the dendritic shaft. The RC mutation impairs the interaction of PKARII β with LRRK2, resulting in increased translocation of PKARII β in the dendritic spines, leading to increased synaptic PKA activities ^[16]. How pharmacological inhibition of LRRK2 kinase activity would affect LRRK2-PKA interactions is difficult to predict. Although there is evidence of crosstalk between LRRK2 and PKA, with PKA acting both upstream and downstream of LRRK2 ^[18, 19]; the precise signaling mechanism requires further investigation. Our behavioral data with MLI-2 treated mice suggests a LRRK2 kinase-dependent mechanism that accounts for the motor impairment in the RC mice.

Based on the reversal of motor impairment after MLI-2 administration, one explanation for the specificity in the impaired response of RC mice could be a higher LRRK2 kinase activity rendered by RC mutation,

compared to GS mutation after the mice underwent the motor learning experiment. This has been suggested previously at basal levels in other tissues or cell lines ^[20, 21]. Thus, we decided to assess the kinase activity of LRRK2 in the striatal extracts of mice trained in specific striatal motor learning.

We relied on p-S1292 LRRK2 (an established *in vivo* LRRK2 autophosphorylation site ^[22]) as a comparative readout for LRRK2 kinase activity across mutations after the end of the behavioral task. Our Western blot analyses showed that the S1292 autophosphorylation of LRRK2 is increased only in GS and not RC striatal extracts. This is consistent with a published study showing increased S1292 only in GS and not in RC striatal extracts at basal levels ^[23].

Additionally, a recent study showed that phosphorylation of S106-Rab12 is an accurate measure of LRRK2 kinase activity in both the periphery and the brain. Importantly, phosphorylation of S106-Rab12 is reversed by the MLI-2 LRRK2 kinase inhibitor in the GS mice ^[24], indicating decreased LRRK2 kinase activity. Accordingly, we examined whether pS106-Rab12 could be a reliable kinase activity marker in the striatal extracts across mutations. In agreement with the literature, we show an increase in the pS106-Rab12 in the GS mice but there is no increase in the RC mice (**Figure R1a**), suggesting that S106-Rab12 is not a generalizable marker of LRRK2 kinase activity. In addition, we showed that the LRRK2 dependent phosphorylation of T73 of Rab10 is increased in the lung of RC and not GS mice consistent with an earlier publication ^[23]; however, we were not able to observe consistent changes in p-73 Rab10 in the striatum extracts of both GS and RC mice at basal levels (**Figure R1b**). Our findings are in complete agreement with three earlier studies in both rat and mouse brains, indicating that although pT73-Rab10 is an excellent readout of LRRK2 kinase activity in peripheral tissues, it does not serve as a sensitive readout in the rodent brain ^[23-25]. Overall, our and others' findings agree that there is no reliable readout of LRRK2 kinase activity in the brain to be used comparatively across LRRK2 mutations.

Figure R1: Western blot analysis of striatal (a), and lung and striatal (b) extracts of WT, RC, and GS mice, probed for the indicated antibodies.

In agreement with the Reviewer's comment, given the critical role of A2A receptor-dependent PKA signaling in affecting motor learning ^[17, 26], altered PKA signaling in the RC mice likely explain the striatal motor learning impairments specific to the RC LRRK2 mutation after dopamine receptors antagonism. Therefore, we decided to focus on that in the revised manuscript. In support of this idea, our new quantitative mass spectrometry data from WT, RC, and GS synaptosomal striatal extracts of mice performing the striatal motor learning test, showed significant increases in the cAMP signaling pathway in the RC and not GS mice

compared to WT (**Suppl. fig. 7**). Our Western blot analyses confirmed this behaviorally relevant increase in the PKA signaling in RC but not GS synaptosomal striatal fractions (**Figure 4**).

Our new findings suggest that balanced PKA signaling is critical for striatal motor learning (**Figure 5e & f**). Specifically, to provide a direct link between PKA activities and impaired performance in the RC mice, we used an adeno-associated virus (AAV)-shRNA (AAV-eGFP-sh m-PKARII β) to knock down the gene encoding the PKARII β —the predominant isoform of PKA in the striatum^[27]. We injected a subset of RC mice with either AAV-eGFP-sh m-PKARII β or AAV-eGFP-sh control (**Figure 5**). The AAV-eGFP-sh m-PKARII β administration resulted in decreased PKARII β levels, compared to AAV-eGFP-sh control. As expected, knockdown of the PKARII β gene led to decreased PKARII β levels and decreased PKA activity in the striatum (**Figure 5**), as PKARII β is the predominant regulatory subunit that drives PKA striatal activity. More importantly, the RC mice injected with the AAV-eGFP-sh m-PKARII β but not AAV-eGFP-sh control showed a gradual behavioral improvement (reversal of their motor learning impairment) similar to that observed in WT mice (**Figure 5c & d**).

In summary, there is not a consistent readout of LRRK2 kinase activity that can be generalized across mutations in the brain. Evaluating differences in kinase activity and how MLI-2 application in the striatum would affect such differences is not currently possible. However, we provide strong evidence in this revised manuscript that the observed differential response of RC in the dopamine antagonism of striatal motor learning is PKA-dependent.

In Fig. 2, other measures of intrinsic excitability such as resting membrane potential, input resistance, rheobase, etc. would strengthen the paper and perhaps shed light on underlying mechanism of LRRK2-RC mutant's reduced iSPN intrinsic excitability.

Response 1.4. We have now included an expanded analysis of intrinsic properties as shown in **Suppl. figs. 1–3** and **Table 4 & 5**. There were no changes in the input resistance or resting membrane potential of SPNs in either RC or GS mice. However, there was an increased rheobase in the SPNs of GS mice but no change in SPNs of RC mice. We further expanded our study to include a host of general membrane properties (**Suppl. fig. 2** and **Table 4**) and action potential properties (**Suppl. fig. 3** and **Table 5**).

Overall, the expanded analysis of membrane properties did not provide obvious mechanistic insights that would explain the observed deficits of dopamine release in both RC and GS mutants, the hypoexcitability in RC iSPNs, and the learning impairment in RC mice. We, therefore, followed the suggestion of the reviewers to examine the role of PKA for a functional insight (see **Response 1.2–1.3**). This revised, expanded work builds on our previous report that dopamine activated LRRK2 negatively regulates PKA activity^[11]

Control for the nonspecific effect of MLI-2 would add strength to the conclusion about kinase activity of LRRK2 as a potential mechanism for motor learning deficit. Does the inhibitor restore the pLRRK2 levels in LRRK2-RC mutants back to WT levels? (Fig. 3d).

Response 1.5. To examine the effect of the MLI-2 inhibitor administration on the mice's performance in the rotarod paradigm (**Figure 3**), we administered 5 mg/Kg MLI-2 daily, 60 min prior to training. MLI-2 is a potent,

selective LRRK2 inhibitor that can cross the blood barrier ^[28, 29]. We report no changes in the performance of MLI-2 treated mice compared to saline treated ones (**Figure R2**). These data suggest that at least in this specific behavioral task, the application of the LRRK2 kinase inhibitor itself does not result in alterations in the mice's performance.

Based on our previous data and Reviewers' suggestions, we now provide evidence that increased PKA signaling underlies the specificity of RC mice in the motor learning deficit (**Figures 4 and 5**). Given our new data and since there is no reliable kinase activity marker for the RC activity in the brain as explained in **Response 1.3**, we did not pursue the altered LRRK2 activity across mutations direction further in the revised manuscript.

Figure R2: MLI-2 administration did not affect the motor performance of WT and RC mice in the rotarod task. WT and RC mice were administered 5mg/Kg MLI-2 daily, 60 minutes prior to testing, throughout the behavioral paradigm. While there was a time main effect $p < 0.0001$ for both WT and RC mice, no treatment effect was observed in either (two way ANOVA, $n_{WT_saline} = 16$, $n_{WT_MLi-2} = 13$, $n_{RC_saline} = 20$, $n_{RC_MLi-2} = 13$).

RC mice have deficient ability to recover motor learning after treatment with a mixture of D1 and D2 antagonists whereas GC has the same recovery as WT mice (Fig. 3). On the other hand, both RC and GS show deficit when treated with D2 antagonist (Fig. 3 Supplementary). In fact, the deficit is more apparent in GS mice. What is the interpretation on this discrepancy?

Response 1.6 We agree with the Reviewer that the GS mice appear to have a transient lower performance in the initial no-drug phase (**Suppl. fig. 5c**). However, our analysis of the time course data did not suggest a bigger deficit in the GS-D2 group as we did not find any significant differences between GS-D2 and WT- or RC-D2 in the session by session *post hoc* tests. In addition, by analyzing averaged latency, regardless of genotype, the latency to fall in all mice that received the D2 antagonist was not significantly different compared to their corresponding saline groups. It is perhaps not surprising that GS shows a transient (non significant) vulnerability to D2 disruption as there is evidence of subtle D2 receptor dependent nigrostriatal ^[30] and glutamatergic ^[31] synaptic dysfunctions of GS mice. Overall, our statistical analysis shows that GS mice

recovered by the end of the behavioral task regardless of whether they received D1+D2 cocktail or D2 antagonist alone.

They note only N for recordings, but should note N for mouse # as well.

Response 1.7. We included all the n's for all the mice in the legends. In addition, we have now also included **Tables 4 & 5** that summarize the data and include all n's for all measures and mice for easier access to readers.

Fig. 3 Supp c, d showed early deficit in RC and GS, not late sessions. Inspection of 3c shows that differences more obvious in sessions 8 to 12 in-between early and late. What is the rationale for dividing them into these time periods? Is there overall time effect?

Response 1.8. We agree with the Reviewer that there are seemingly group differences in the middle of the no-drug phase. Previous studies have already established that the initial degraded performance of WT mice in the drug phase improves only gradually ^[17, 32]. In the revised manuscript, we confirmed that there is a main effect of time and a significant three-way interaction (genotype by treatment by time) in the time course data. Following this significant interaction with a session by session, *post hoc* group comparisons did not reveal any differences between genotypes in the middle of the no-drug phase. However, we did find differences towards the end of the session. As our main goal in the present study was to uncover any differences in the performance of mutant LRRK2 KI mice, we decided to focus on early and late sessions in the behavioral task. Why do we present the data as an averaged latency to fall? There is the possibility that the unusually large number of *post hoc* comparisons for the time course data may generate type II errors, which could be avoided by analyzing the averaged latency to reduce the number of pairwise comparisons. Secondly, we think that averaged latency may even out the day-to-day fluctuations in animals' status and better reflect their performance.

Technical points, typo etc.

1) Line 112, "mice was administered with 5 mg/kg MLI-2 (LRRK2 inhibitor)". What was MLI-2 dissolved in – saline or DMSO?

Response 1.9. The MLI-2 was first dissolved in DMSO (25 mg/ml) and then diluted in saline with ultrasonic dissociation. The working solution contained 5% DMSO. This has now been updated in the Materials and Methods section.

2) Line 113-114, "All the injections were given i.p (0.1 ml/kg)". This seems to translate to giving a 20 g mouse an IP injection of 0.002 ml = 2 ul, which seems quite low. Please double check

Response 1.10. That was a typographical error. We have now corrected it to 0.005 ml/g.

3) Line 209 "SCHSCH23390": should be SCH23390?

Response 1.11. That has been corrected.

4) Line 232-234, "Therefore, we treated RC mice with the LRRK2 inhibitor MLI-2 prior to D1 and D2 receptor antagonism. We found motor improvement across behavioral sessions during the drug-free phase of MLI-2-treated mice...". It is unclear through the paper if mice were treated with MLI-2 only for sessions 1-5 (as seems to be suggested by the above sentences), or if mice were treated with MLI-2 during sessions 1-18, as seems to be suggested by the method section that MLI-2 was given "prior to all daily sessions" (line 112). The interpretation of LRRK2's role in rotarod learning vs. performance obviously depends critically on clearing this up.

Response 1.12. The MLI-2 was administered throughout sessions 1 through 18. This is now clearly stated in the manuscript.

5) Lines 260-261, "In contrast, RC mice did not exhibit the performance of their saline controls across sessions". However, the corresponding P value from Fig. 3 table supplement 1 (Session 16-18, RC, saline vs D2 antagonism) seems to be $P = 0.1492$?

Response 1.13. The Reviewer is right, as there was no change in the performance in the last three sessions found across genotypes (**Table 7**). The WT and GS mice treated with D2 receptor antagonist reached the performance of their saline controls at the last session of the task (session 18: $\text{latency}_{\text{GS_eticlopride}} = 111.9 \pm 9.1$ s, $\text{latency}_{\text{GS_saline}} = 139.1 \pm 10.5$ s; $p = 0.5789$, $\text{latency}_{\text{WT_eticlopride}} = 135.2 \pm 23.1$ s, $\text{latency}_{\text{WT_saline}} = 133.8 \pm 15.2$ s; $p = 0.9999$, 2-way ANOVA, Sidak's *post hoc* tests). In contrast, RC mice did not exhibit the performance of their saline controls for the last studied session ($\text{latency}_{\text{RC_eticlopride}} = 106.3 \pm 14.3$ s, $\text{latency}_{\text{RC_saline}} = 173.3 \pm 15.2$ s $p = 0.0042$; 2-way ANOVA with Sidak's *post hoc* tests).

The text has been updated to reflect that.

6) Figure 3 table supplement 1: "D1 antagonism" column title was repeated twice. One of them probably should be "D2 antagonism".

Response 1.14. This is now fixed.

Reviewer #2 (Remarks to the Author):

Regrets for the delay in returning the review.

Key findings in the manuscript by Xenias et al:

The purpose of the study is to identify electrophysiological differences in dopamine-related neurotransmission (and behavior) in *Lrrk2* KI mice as models of late-onset LRRK2-linked PD. The study is interesting and technically

appears to be well done. Principal findings comprise:

1 - The paper demonstrates that p.R1441C and p.G2019S knock-in mutations in Lrrk2 are responsible for a decrease in nigrostriatal dopamine release;

2 - Based on that finding the expectation that the authors had is that there would be differences in excitability of striatal projection neurons (SPN). To interrogate direct- and indirect-pathways, they tested the number of spikes in related projection neurons, i.e., dSPNs and iSPNs.

The most puzzling finding of their work is that the number of spikes were lower only in p.R1441C but not p.G2019S mutant animals, and only in the iSPN category, but not in the dSPN-linked pathway. Nevertheless, there was a trend in p.G2019S mice;

Response 2.1. While visually there was a very slight reduction of excitability of GS iSPNs when compared to WT littermates, there was no significant difference. Only RC iSPNs showed a significant reduction in excitability. As reduced nigrostriatal dopamine is associated with reduction of iSPN excitability and corticostriatal synaptic input ^[7], the non-significant changes in excitability in GS iSPNs could be owed to the reduction of dopamine release in GS and suggests that an additional change is required to explain the significant reduction of excitability of RC iSPNs. In our revised work, we show that there is an increase in striatal PKA activity unique to RC mice that underlies the impairments to motor learning. The distinct postsynaptic signaling pathways resulting from increased PKA activity in the RC mice might contribute to the decreased excitability in the RC iSPNs. For a further and detailed discussion of the function of PKA (and the behavioral significance we have since determined), intrinsic electrophysiological properties (which have been greatly expanded in this revision), and the results with MLI-2 in the context of PKA signaling, **please also see Responses 1.3 –1.5.**

3 - Increased membrane capacitance was found for iSPNs in p.R1441C mice, but not in p.G2019S mice;

4 - No difference in paired-pulse ratios (PPRs) but increased excitatory postsynaptic current (EPSC) decay time was observed in then iSPN pathway of p.R1441C mice;

5 - The authors suggest that Lrrk2 kinase activity "we treated RC mice with the LRRK2 inhibitor MLI-2 [33] prior to D1 and D2 receptor antagonism" is the reason that underlies the differences seen vs. WT in the rotarod performance.

Comments and suggestions for edits / improvements:

1 – In their introduction, the authors did not mention other KI mice-based, similar investigations and related findings that had already been published by other authors, such as their (later) references #20 and #21;

Response 2.2. We have now expanded our introduction to include the references suggested by the Reviewer, and we attempted to both synthesize existing knowledge and state our hypotheses that drove our study more clearly. Please see **Response 2.3.**

2 - A more refined intro would also lend itself to a specific hypothesis statement for the proposed work, which was sadly missing;

Response 2.3. We have now rewritten the introduction per the Reviewer's suggestion. Please also see **Response 2.2.**

3 - In their KI mice, have the authors convinced themselves that the two KI lines, i.e., GS and RC, indeed have the same mRNA and protein levels as do WT mice? This seems to be important given their different outcomes in electrophysiological tests. Herzig et al in 2011 have shown how distinct KI mutations can alter steady-states in different tissues;

Response 2.4. To address the Reviewer's comment, we have performed Western blot analysis of LRRK2 proteins and showed no difference in the protein levels across WT, RC, and GS mice. The representative Western blot and the quantification are shown in **Suppl. fig. 6.**

4 – Was there a sex effect in fast scanning cyclic voltammetry? How many mice of each sex and what ages were used in their studies? The effect of sex in the context of some *Lrrk2*-linked phenotypes in mice has been published recently.

Response 2.5. We have expanded our voltammetric analysis to sex-balance both the wildtype and mutant datasets. While it appeared there was a larger variability of dopamine release in males across all genotypes, there were no significant differences between the sexes for any of the groups. These data are presented as scatter plots in **Figure R3**. A table summarizing the data, sample sizes, and sex are now included (see **Table 1**). All mice used in our ex vivo studies were from P90–P110. This information is more clearly stated in the Methods section.

Figure R3: Evoked dopamine release from WT, RC, and GS mice shown by sex. While there was a difference for each mutant compared to WT, there were no differences between sex for any of the groups ($n_{WT_males} = 7$, $n_{WT_females} = 3$, $n_{RC_males} = 4$, $n_{RC_females} = 5$, $n_{GS_males} = 5$, $n_{GS_females} = 8$).

5 – Given the availability of antibodies now, it seems that measuring the levels of distinct p-Rab proteins in components of both the direct and indirect pathway from WT, RC and GS seems to be the logical next step to do in order to further elucidate the underlying mechanisms. Short of that a simple explanation for the interesting observation the authors made remains unanswered;

Response 2.6. Please see **Response 1.3** regarding the use of p-Rab10 antibodies as a sensitive readout of LRRK2 kinase activity in the brain. The S1292 autophosphorylation as an alternative readout is increased only in the GS but not RC mice. There was not a consistent readout that allowed us to evaluate the activity of LRRK2 kinase across mutations in the brain.

6 – In addition, could the authors provide evidence of the PKA pathway involvement in the p.R1441C driven outcomes as mentioned in the Discussion, in order to further enrich the mechanistic insight?

Response 2.7. We have now substantially expanded our work to directly examine the role of differential signaling of PKA between RC and GS mice. This includes two new **Figures 4–5** and **Suppl. fig. 7** and a completely revised, extensive discussion relating to the observed increased levels of PKA activities specific to RC mice and its contribution to the observed phenotypes.

7 - The experiments are done in heterozygous animals, which is laudable to mimic the human genotypic findings. The question arises: Does homozygosity of either RC or GS increase the measured outcomes (i.e., dopamine release, number of spikes, capacitance and rotarod performance)?

Response 2.8. We focused on heterozygotes LRRK2 KI mice since LRRK2 GS and RC mutations are Mendelian dominant ^[33] and heterozygotes LRRK2 mutation carriers exhibit clinically similar manifestations as homozygotes ^[34]. Although it is possible that there are quantitative differences in the outcomes, both heterozygous and homozygous GS KI mice had the same key phenotypes in relation to PKA localization and signaling ^[11]. Moreover, our previous findings showed that heterozygous RC KI mice respond similarly to homozygous mice in regards to PKA regulation ^[11]. For these reasons, we elected not to perform additional experiments in homozygous mice. We have now emphasized our rationales in the Discussion.

8 - Was the MLI-2 dissolved in saline or DMSO? If DMSO was used, figure 3 might benefit from having DMSO control. DMSO can change gene expression, see PMID: 30646511. Also, the legend for figure 3 should state the actual amount of administered MLI-2;

Response 2.9. MLI-2 was dissolved in 5% DMSO. We agree with the Reviewer that the 5% DMSO would be the appropriate control. Given the extremely large number of mice required for comparing three genotypes, undergoing four different pharmacological treatments, along with two different viral constructs injections, we used saline as the universal control. We report that MLI-2 resulted in no changes in the performance (**Figure R2**). This strongly argues that the small percentage of DMSO has no significant impact on the behavioral phenotype of these mice. We have acknowledged this comparison in the revised manuscript. In addition, the new **Suppl. fig. 5** legend now states the amount of MLI-2 administered per the Reviewer's suggestion. Please also see **Response 1.9**

9 - At the MLI-2 dose used in the rotarod experiment, do the authors have knowledge of off-target effects by MLI-2 ? From where was MLI-2 procured?

Response 2.10. There has been a detailed characterization of the MLI-2 inhibitor in the literature. Besides its clear potential to reduce cytotoxicity associated with LRRK2 mutations in PD-relevant cell and animal models ^[35] MLI-2 has been found to be ~300x more selective for LRRK2 over other kinases and can readily cross the blood-brain barrier in mice ^[28]. Its potency and efficacy were recently further confirmed in non-human primates ^[29]. In this study, we used an acute, low dosage treatment that is shown to be efficacious but with little off-

target effects. While chronic use (10 weeks, 60 mg/Kg) ^[24] revealed some subtle alterations in the biochemical signature of peripheral tissues involving the endolysosomal and mitochondrial system in recent proteomics data, no functional and morphological impairments across tissues were reported. Importantly, we showed that the administration of the MLI-2 inhibitor had no effect on the performance of both WT and RC in the rotarod task (**Figure R2**). The information of where the inhibitor is procured has been added to the Materials and Methods section.

10 – The rationale for the usage of MLI-2 in p.R1441C mice treated with D1+D2 antagonist could be better explained;

Response 2.11. It is accepted that LRRK2 mutations lead to a gain of function of kinase activity ^[36]. Therefore, we decided to suppress the LRRK2 kinase activity, using a potent, highly selective LRRK2 kinase inhibitor to examine if hyperactive LRRK2 kinase contributed to the motor deficits effects we observed in the RC mice. Interestingly, the RC mutation, mainly in the context of heterologous cell lines, leads to a higher increase in Rab phosphorylation compared to the GS mutation ^[20], which is localized to the kinase domain. Therefore, we hypothesized that if RC resulted in even higher kinase activity compared to the GS, this could explain the specific degraded performance in the RC vs GS mice in the rotarod task. However, in the light of the differential effects in LRRK2 kinase activity observed across mutations (please see **Response 1.3** and **Figure R1**), our data can not support this scenario. Instead, following all Reviewers' suggestions, we now decided to focus this study on the selective effect of RC mice in the PKA signaling as the mechanism underlying the impaired motor learning. In particular, we showed that in RC mutants, PKA activity is increased in the striatal synaptosome and importantly that the learning deficits in RC mice can be reversed with PKA shRNA knockdown. Our expanded work in this revision now revealed an underlying mechanism through PKA, whose dysregulation by RC LRRK2 accounts for the learning deficits unique to RC mice.

11 – Other controls come to mind, for example, measuring dopamine release with and without MLI-2 (i.e., vehicle control), measuring the number of spikes in iSPNs with and without MLI-2, measuring capacitance of iSPNs with and without MLI-2.

Response 2.12. We agree with the Reviewer regarding the importance of further investigation on how LRRK2 inhibitors would influence dopamine release. However, in the revised manuscript we put our efforts on uncovering a molecular mechanism that was well motivated for the observed differential phenotype changes between the LRRK2 mutants. In short, we show that p-PKA activities were selectively increased in RC mice. The knowledge gained should inform specific LRRK2 domain targeting of future LRRK2 inhibitors to test for the differential phenotypic changes we report. Please see also **Response 1.3**

Reviewer #3 (Remarks to the Author):

Xenias and co-authors investigate the effect of two distinct LRRK2 mutations, G2019S and R1441C, in dopaminergic neurotransmission. Using electrophysiological and behavioral analyses in knockin mice, they found that while both mutations cause dopaminergic deficits, they likely operate through different mechanisms. In particular, they observe that LRRK2-R1441C exhibits reduced nigrostriatal dopamine release in the indirect pathway which is accompanied by impaired motor learning. Administration of LRRK2 inhibitors rescues the

behavioral deficits associated with DA-receptor antagonism in R1441C mice. Instead, despite the observed impairment in DA release in G2019S KI mice, no behavioral or electrophysiological alterations were found, at least

in the analyses performed.

Although there are novel findings in this study related to the R1441C mutation, overall there is not substantial novelty and a comprehensive investigation of the molecular mechanisms underlying the observed phenotypes is missing.

I have a number of specific comments/suggestions to strengthen this study:

- Using fast scanning cyclic voltammetry, the authors found decreased evoked DA release in RC and GS slices, indicating that the DA signaling is impaired, as previously suggested at least for the G2019S mutation. At this point, to test at what level the impairment is, the authors should evaluate possible differences in DA signaling components, eg D1 and D2, pPKA/PKA, DARPP32, pERK1/2/ERK1/2 etc It would be also interesting to measure DAT levels and activity.

Response 3.1. We have followed the Reviewer's suggestion, and in order to identify the level of impairment that underlies the observed phenotypes, we have performed Western blot analysis in striatal synaptosomal fractions of WT, RC, and GS mice. While we observed no differences in D1, D2, and DAT levels across genotypes, we found an increase in the phosphorylation of T34-DARPP32, specifically in the RC and not GS mice (**Suppl. fig. 6**). As phosphorylation of T34-DARPP32 is PKA mediated, these findings are consistent with our recent study showing an increase in PKA activities in the RC and not GS mice ^[16]. Our findings suggest that at least aberrant PKA signaling within the striatum might contribute to the observed alterations in the RC mice. These data are now part of the new **Suppl. fig. 6**.

- The AP recording graphs are unclear. There are no reference of the half-maximum firing in the graphs, eg in figure 2b the half-maximum firing rate does not correspond to 625 pA.

Response 3.2 We thank the Reviewer for pointing out the confusion regarding the current shown in **Figure 2b**. This current ($I = 625$ pA) corresponds to the maximal difference in firing between the WT and RC iSPNs. We have now provided analysis on the half-maximal changes to quantify further the changes in firing were in the iSPNs between WT and RC mice. Other than significant changes in the firing of iSPNs in RC mice, no other changes were observed. The Results section covering these points has been rewritten to better clarify the points of these two different analyses.

- To investigate whether the differences in excitability are due to differences in membrane size, they measured membrane capacitance and found increased C in the iSPNs of RC mice. Do these neurons present differences in arborization/spine number or maturation?

Response 3.3 While the increased capacitance of iSPNS in RC mice is in agreement with the hypoexcitability of iSPNs observed, this correlation of changes would not reveal the causation of the alterations and why we didn't pursue further doing a full reconstruction of the dendritic architecture. Moreover, we previously investigated changes in spine number and maturation and reported none ^[16]. In line with the Reviewer's

suggestions, we instead focused on pursuing the functional role of PKA in these LRRK2 mutants that would account for the changes observed (please see **Response 1.2—1.3**)

-Since they register in current clump, why not evaluating Na⁺ and Ca²⁺ currents, being those influencing AP?

Response 3.4 One of the main focuses of our study was to examine the intrinsic properties of SPNs in LRRK2 mutant mice; iSPNs of RC mice were the only cells that showed an alteration in excitability (i.e., hypoexcitable) (**Figure 2**). However, as the action potential threshold and waveform were unaltered (**Table 5**), it is unlikely these changes were attributable to the altered properties of Na⁺ channels. How Ca²⁺ currents are associated with firing properties in SPNs is less clear. For these reasons, we elected not to pursue measuring Na⁺ and Ca²⁺ currents in this revision. To provide a more rigorous analysis of the intrinsic properties of SPNs, we have now expanded our study of general membrane properties as well as action potential characteristics (**Suppl. Fig 2–3 and Table 4 & 5**).

Why the capacitance of WT mice is different in supplemental figure 2A versus B? Are they using littermates?

Response 3.5 The capacitances shown for the WT cases for RC and GS (now **Suppl. fig. 2**), were from littermates. In fact, all our *ex vivo* studies were performed on littermates; this is now clarified in the Methods section. Importantly, there were no significant differences in the capacitances of RC versus GS control mice. The apparent differences in the WT capacitances between RC and GS mice are best explained as sampling variability. Additionally, we found that neither the input nor membrane resistance of SPNs for any of the mutants changed (**Table 4**). Lastly, we present in this revision an expanded study of both passive membrane and active membrane properties (please see **Suppl. fig. 2 & 3; Table 4 & 5**).

- In addition of measuring evoked EPSCs, they should also investigate the spontaneous currents. Differences in spontaneous activity would support that the observed differences are postsynaptic.

Response 3.6. Evoked EPSCs were recorded to probe for changes in corticostriatal input. This analysis did not result in information that suggests changes in either pre- or postsynaptic properties of the synapse in either RC or GS mice. In addition, as the cellular origin of spontaneous events is unknown (i.e., cortical vs. thalamic), the results obtained can be difficult to interpret. Accordingly, we did not record spontaneous EPSCs in this revised study.

- Behavior. Authors need to explain why they chose such an intense training (18 trials per day). How can they rule out that the observed differences at the rotarod are not due to differences in cerebellar activity?

Response 3.7. It is established that dopamine loss, besides its direct effect on motor performance, leads to impaired motor learning that degrades future motor performance even after the restoration of dopamine signaling on a rotarod motor-learning task ^[17, 26]. Previous reports unmasked this impaired motor learning in WT mice employing the same accelerating rotarod task we used here accompanied by a reversible D1R and D2R antagonism ^[17, 26]. Despite dopamine release deficits in LRRK2 mutant mice, we and others observed no overt deficits in motor learning at basal conditions. Therefore, we were interested in defining these

mice's striatal motor learning deficits using this previously established rotarod paradigm ^[17]. We agree with the Reviewer that the cerebellum is important for motor learning. However, the specific behavioral paradigm captures how an acquired learning deficit as a result of dopamine D1 and D2 receptor (expressed predominantly in the SPNs of the striatum) antagonism affects future performance and learning in the accelerated rotarod. Consistent with an earlier study [27], our revised manuscript shows the dependency of this motor task on the PKARII β (Figure 5), which is highly enriched in the striatum and is almost undetectable in the cerebellum. The dependence of the specific behavioral test on PKARII β activities, evidenced by our striatal targeted viral-mediated approaches (**Figure 5**), suggests that cerebellar defects are unlikely to contribute to the observed phenotypes. We include these comments now in our Discussion.

- Mli2 treatment of R1441C mice before D1/D2 antagonism rescues the motor learning deficiencies. The rationale of using Mli2 is that R1441C has increased kinase activity. Can they show that the striata of these mice have more kinase activity (pS1292, pRab10)? As the G2019S are also hyperactive, what does Mli2 treatment do in the G2019S mice in the same type of test? And in WT?

Response 3.8. Please see **Response 1.2–1.3**.

- Using D1 and D2 antagonists separately the authors state that the effect depends on D2 but I don't see any statistical differences in the RC mice upon treatment (see Suppl. fig. 3 C & D). How do the authors interpret the delayed response in motor learning after administration of D1/D2 receptor antagonists in R1441C mice?

Response 3.9. Regardless of genotype, D1R and D2R antagonism resulted in an initial degraded performance in the drug-free phase followed by a gradual improvement across sessions. D1R antagonism resulted in a degraded motor performance (i.e., the first 5 days) followed by an immediate recovery upon cessation of the treatment. On the other hand, antagonism of D2R induced abnormal learning that results in a persistent impairment and a slowed recovery in the drug-free phase in all three genotypes. This differential pattern of recovery, depending on D1 or D2 blockade, points to a D2-mediated pattern similarly to what was previously reported ^[17]. In addition, for the last session of the test in the presence of the D2R antagonist, RC mice displayed a significant degrade in the performance compared to WT and GS. Overall, the impaired performance of RC mice is stronger in the presence of both D1 and D2 antagonists, suggesting a synergistic effect of the two receptor subtypes. We have now discussed these points in the discussion session of our revised manuscript.

- In addition of using D1/D2 antagonists, DA receptor agonists should also be used. At low concentrations, pramipexole can fairly discriminate between D1 and D2.

Response 3.10. The purpose of the behavioral paradigm was to study how experience-dependent learning deficit induced by pharmacological blockade of dopamine signaling, previously described in WT mice ^[17], determines future motor performance and learning across LRRK2 mutations. We thus focused on blocking and not activating receptors by performing mixed D1R and D2R antagonism. This was followed by experiments

with blockade with individual receptor subtypes. As the goal was to block and not activate dopamine receptors, we focused on receptor antagonists.

- Unexpectedly there are no differences in the total distance travelled upon D1/D2 receptor antagonists in any genotype (see supplemental figure 3f)? How can this be explained?

Response 3.11. In light of our data showing increased impairment of RC mice in the striatal motor learning, we wanted to test if the administration of dopamine antagonists along with rotarod training impairs animals on other tasks. A subset of mice (after rotarod training and antagonists administration) were tested in the open field test following the 72-hr break. No differences were shown across genotypes in the antagonist-treated, trained mice. These results are consistent with previous studies using the same paradigm in WT mice ^[17]. It is not surprising that the effects of the drugs had worn off after the 72-hr washout period. However, we have now included new data showing that acute D1 and D2 receptor antagonism, 2 hours prior to open field substantially reduced open-field activity (**Figure 3e**). These data suggest that despite the acute effect on motor performance, the dopamine receptor antagonism leading to an impeded experience-dependent learning impairs specifically the subsequent striatal motor learning performance/recovery even after restoring dopamine signaling.

- Since the behavioural experiment suggests learning impairment in RC mice, it would be interesting to add a memory test (novel object recognition)

Response 3.12. The experiment we used is a particular behavioral paradigm to test how experience-dependent impaired learning during the acquisition of the skill in an accelerating rotarod as previously shown in WT mice, would further impede future performance. The approach has been established previously to be related to striatal motor learning ^[17]. As inappropriate corticostriatal plasticity acquired under reduced dopamine is proposed to underlie these processes, a memory test such as novel object recognition likely will not capture corticostriatal alterations.

Reviewer #4 (Remarks to the Author):

In this study, Xenias et al examined dopamine transmission in the R1441C and G2019S LRRK2 KI mouse models. They found a decrease in nigrostriatal dopamine release in both lines. Furthermore, they detected a decrease in excitability of indirect-pathway striatal projection neurons in the R1441C mice, which was accompanied by impairments in dopamine-dependent striatal motor learning. Motor learning deficits were rescued by pharmacological LRRK2 kinase inhibition.

This work confirms previous data showing reduction of dopamine release in G2019S LRRK2 KI mice (PMID 25836420). Using R1441C KI mice used in this study, Tong et al were not able to detect significant changes in striatal dopamine levels (PMID: 19667187). However, Xenias et al employed fast-scanning cyclic voltammetry, which revealed a significant decrease in DA level.

Overall, the findings are interesting. Importantly, instead of focusing on one mutation or the other, this study undertook a systematic comparison in two different mouse lines.

- It is unclear why only the R1441C mice develop LRRK2 kinase-dependent motor impairment.

Response 4.1. The reversal of motor impairments after MLI-2 administration in the RC mice suggests that LRRK2 kinase activity mediated alterations in key signaling pathways determine the dopamine-dependent striatal motor learning in these mice. Should this be the case, the GS mice (where LRRK2 kinase activity is also elevated) were expected to have similar deficits. A possible explanation for motor impairment in RC mice is a greater increase in LRRK2 kinase activity, compared to the GS mice during the striatal motor learning task. Greater kinase activity was previously observed in RC mutation compared to the GS in heterologous cell lines^[20]. Given the differential effect of mutations in the available readouts of LRRK2 kinase activity in the brain (please see **Response 1.31**), we could not compare kinase activity between RC and GS that would explain the RC-specific motor impairment.

On the other hand, PKA signaling is a key signaling pathway underlying striatal motor learning. In the revised manuscript, by employing biochemical, behavioral, and proteomic approaches (**Figures 4 & 5**), we show that synaptic PKA activities are specifically elevated in the SPNs of the RC mice but not in GS mice. Moreover, the manipulation (i.e., decrease) of PKA activities using viral approaches in the RC mice was sufficient to restore the motor impairment in RC mice (**Figure 5**).

The authors speculate that the dissociation between impaired DA release and striatal alterations is linked to changes in striatal PKA signaling in the R1441C mice (line 286). It would be useful to discuss a bit more on the potential role of PKA in this context.

Response 4.2. Based on all Reviewers' suggestions, we generated new data connecting increased PKA signaling in RC mice with the motor impairments in the accelerating rotarod. Specifically, we show that the decrease of the aberrantly elevated p-PKA levels in the RC mice is sufficient to reverse the learning impairment in the rotarod task (**Figures 4 & 5** and **Suppl. fig. 7**). We have now rewritten the manuscript with a focus on the dysregulation of postsynaptic PKA signaling specific to RC mice.

The authors conclude "our data argue that the impact of LRRK2 mutations cannot be simply generalized". While it is already known that the impact of these two mutations is different, the authors should discuss more on the specific effects that they may have on different pathways and how this could provide opportunities for targeted therapies.

Response 4.3. We have now expanded the Discussion, based on the Reviewers' suggestions.

References

1. Lahiri, A.K. and M.D. Bevan, Dopaminergic Transmission Rapidly and Persistently Enhances Excitability of D1 Receptor-Expressing Striatal Projection Neurons. *Neuron*, 2020.
2. Ericsson, J., et al., dopamine differentially modulates the excitability of striatal neurons of the direct and indirect pathways in lamprey. *J Neurosci*, 2013. 33(18): p. 8045-54.
3. Ding, L. and D.J. Perkel, Dopamine modulates excitability of spiny neurons in the avian basal ganglia. *J Neurosci*, 2002. 22(12): p. 5210-8.
4. Shen, W., et al., Dichotomous dopaminergic control of striatal synaptic plasticity. *Science*, 2008. 321(5890): p. 848-51.
5. Kreitzer, A.C. and R.C. Malenka, Dopamine modulation of state-dependent endocannabinoid release and long-term depression in the striatum. *J Neurosci*, 2005. 25(45): p. 10537-45.
6. Wu, Y.W., et al., Input- and cell-type-specific endocannabinoid-dependent LTD in the striatum. *Cell Rep*, 2015. 10(1): p. 75-87.
7. Lee, S.J., et al., Cell-type-specific asynchronous modulation of PKA by dopamine in learning. *Nature*, 2021. 590(7846): p. 451-456.
8. Fieblinger, T., et al., Cell type-specific plasticity of striatal projection neurons in parkinsonism and L-DOPA-induced dyskinesia. *Nat Commun*, 2014. 5: p. 5316.
9. Chan, C.S., et al., Strain-specific regulation of striatal phenotype in *Drd2-eGFP* BAC transgenic mice. *J Neurosci*, 2012. 32(27): p. 9124-32.
10. Day, M., et al., Selective elimination of glutamatergic synapses on striatopallidal neurons in Parkinson disease models. *Nat Neurosci*, 2006. 9(2): p. 251-9.
11. Parisiadou, L., et al., LRRK2 regulates synaptogenesis and dopamine receptor activation through modulation of PKA activity. *Nat Neurosci*, 2014. 17(3): p. 367-76.
12. Baines, R.A., Postsynaptic protein kinase A reduces neuronal excitability in response to increased synaptic excitation in the *Drosophila* CNS. *J Neurosci*, 2003. 23(25): p. 8664-72.
13. Yao, W.D. and C.F. Wu, Distinct roles of CaMKII and PKA in regulation of firing patterns and K(+) currents in *Drosophila* neurons. *J Neurophysiol*, 2001. 85(4): p. 1384-94.
14. Piccoli, G., et al., LRRK2 controls synaptic vesicle storage and mobilization within the recycling pool. *J Neurosci*, 2011. 31(6): p. 2225-37.
15. Shin, N., et al., LRRK2 regulates synaptic vesicle endocytosis. *Exp Cell Res*, 2008. 314(10): p. 2055-65.
16. Chen, C., et al., Pathway-specific dysregulation of striatal excitatory synapses by LRRK2 mutations. *Elife*, 2020. 9.
17. Beeler, J.A., et al., A role for dopamine-mediated learning in the pathophysiology and treatment of Parkinson's disease. *Cell Rep*, 2012. 2(6): p. 1747-61.
18. Muda, K., et al., Parkinson-related LRRK2 mutation R1441C/G/H impairs PKA phosphorylation of LRRK2 and disrupts its interaction with 14-3-3. *Proc Natl Acad Sci U S A*, 2014. 111(1): p. E34-43.
19. Greggio, E., L. Bubacco, and I. Russo, Crosstalk between LRRK2 and PKA: implication for Parkinson's disease? *Biochem Soc Trans*, 2017. 45(1): p. 261-267.
20. Alessi, D.R. and E. Sammler, LRRK2 kinase in Parkinson's disease. *Science*, 2018. 360(6384): p. 36-37.
21. Purlyte, E., et al., Rab29 activation of the Parkinson's disease-associated LRRK2 kinase. *EMBO J*, 2018. 37(1): p. 1-18.
22. Sheng, Z., et al., Ser1292 autophosphorylation is an indicator of LRRK2 kinase activity and contributes to the cellular effects of PD mutations. *Sci Transl Med*, 2012. 4(164): p. 164ra161.
23. Iannotta, L., et al., Divergent Effects of G2019S and R1441C LRRK2 Mutations on LRRK2 and Rab10 Phosphorylations in Mouse Tissues. *Cells*, 2020. 9(11).

24. Kluss, J.H., et al., Preclinical modeling of chronic inhibition of the Parkinson's disease associated kinase LRRK2 reveals altered function of the endolysosomal system in vivo. *Mol Neurodegener*, 2021. 16(1): p. 17.
25. Nguyen, A.P.T., et al., Dopaminergic neurodegeneration induced by Parkinson's disease-linked G2019S LRRK2 is dependent on kinase and GTPase activity. *Proc Natl Acad Sci U S A*, 2020. 117(29): p. 17296-17307.
26. Zhuang, X., P. Mazzoni, and U.J. Kang, The role of neuroplasticity in dopaminergic therapy for Parkinson disease. *Nat Rev Neurol*, 2013. 9(5): p. 248-56.
27. Brandon, E.P., et al., Defective motor behavior and neural gene expression in Rllbeta-protein kinase A mutant mice. *J Neurosci*, 1998. 18(10): p. 3639-49.
28. Fell, M.J., et al., MLI-2, a Potent, Selective, and Centrally Active Compound for Exploring the Therapeutic Potential and Safety of LRRK2 Kinase Inhibition. *J Pharmacol Exp Ther*, 2015. 355(3): p. 397-409.
29. Baptista, M.A.S., et al., LRRK2 inhibitors induce reversible changes in nonhuman primate lungs without measurable pulmonary deficits. *Sci Transl Med*, 2020. 12(540).
30. Volta, M., et al., Initial elevations in glutamate and dopamine neurotransmission decline with age, as does exploratory behavior, in LRRK2 G2019S knock-in mice. *Elife*, 2017. 6.
31. Tozzi, A., et al., Dopamine D2 receptor activation potently inhibits striatal glutamatergic transmission in a G2019S LRRK2 genetic model of Parkinson's disease. *Neurobiol Dis*, 2018. 118: p. 1-8.
32. Beeler, J.A., et al., Dopamine-dependent motor learning: insight into levodopa's long-duration response. *Ann Neurol*, 2010. 67(5): p. 639-47.
33. Paisan-Ruiz, C., et al., Cloning of the gene containing mutations that cause PARK8-linked Parkinson's disease. *Neuron*, 2004. 44(4): p. 595-600.
34. Zimprich, A., et al., The PARK8 locus in autosomal dominant parkinsonism: confirmation of linkage and further delineation of the disease-containing interval. *Am J Hum Genet*, 2004. 74(1): p. 11-9.
35. Hatcher, J.M., et al., Small-Molecule Inhibitors of LRRK2. *Adv Neurobiol*, 2017. 14: p. 241-264.
36. Cookson, M.R., The role of leucine-rich repeat kinase 2 (LRRK2) in Parkinson's disease. *Nat Rev Neurosci*, 2010. 11(12): p. 791-7.

Reviewers' comments:

Reviewer #1 (Remarks to the Author):

The manuscript has been extensively revised with addition of several new data. Notably the authors now provide data on PKA activity increase in RC mutant and show potential mechanisms of reduced excitability in SPN as opposed to GS mutant with similar DA release changes without excitability change. Overall the manuscript provides very interesting and in-depth insights on the effect of LRRK2 mutations on striatal function.

- Results, "Increased synaptic PKA activities in the striatum of R1441C mice following striatal motor learning" section, "The phospho-PKA levels in the striatal synaptosomal preparations were decreased at the end of the task in the WT mice ; at a time point, their initial impaired performance was improved, similar to their saline controls. However, the PKA signaling remained elevated in the RC mice correlating with their impaired performance in the rotarod test". Although the western blot in Fig. 4g seems to agree with this claim, there was no quantification or statistics. These should be included to substantiate the claim that p-PKA substrates decreased for WT from session 5 to 18, but remained both high across time points, and higher than WT, for RC.
- Results, "Aberrant PKA signaling underlies the motor deficits of R1441C mice". Although sh-PKARIIb increased performance of RC mice previously treated with DA receptor antagonist, it had the opposite effects on WT mice and reduced their performance. Could the authors comment on this counterintuitive finding? For instance, does it imply that motor performance isn't a simple monotonic function of p-PKA substrate? Could PKARIIb knock-down in dSPN vs. iSPN play a role in this counterintuitive finding?
- Results, "Aberrant PKA signaling underlies the motor deficits of R1441C mice". In addition to the single western blot shown in Fig. 5b, do the authors have quantification or statistics of PKARIIb or p-PKA substrate levels after sh-control vs. sh-PKARIIb? For instance, did sh-PKARIIb reduce p-PKA substrate in RC mice back to WT's sh-control level, thereby rescuing motor performance? Additionally, did sh-PKARIIb reduce WT's p-PKA substrate to an even lower level (e.g., reference 73, Brandon et al. 1998)? Data like these would suggest that there may be an optimal level of p-PKA substrate for motor performance.

Minor Comments:

- Methods, "Viral-mediated short-hairpin RNA knockdown" section: when was the shRNA AAV injected relative to the rotarod experiment?
- Methods, "Quantitative proteomics and analysis" section, "129 for WT, 130 for RC, and 131 for GS for three biological replicates per the manufacturer's instructions". Please clarify: does this mean 3 biological replicates for each genotype, or 3 replicates in total (1 for each genotype)?
- Fig. 3: there does not seem to be any mention of Fig. 3e in the main text. -What is its purpose? And what is the inset figure in Fig. 3e and its purpose?
- Results: "R1441C mice show impaired dopamine-dependent motor learning" section, "D1 receptor antagonism immediately improved the drug-free phase in all genotypes". The sentence seems potentially confusing? Maybe "Mice previously treated with D1 receptor antagonist showed immediate rotarod improvement after 72 hr washout, regardless of genotype"?
- For synaptosome fraction extraction, both for western blot and proteomic analysis (Fig. 4 and Suppl Fig. 6), there is no information about when the striatum was extracted relative to rotarod training and dopamine receptor antagonist. Was it immediately after 5 sessions of rotarod training with dopamine receptor antagonist treatments? Or was it after the 72 hour washout period? Or some other time point? This can be important because dopamine antagonist itself can have time-dependent effect on the proteome, especially on protein phosphorylation.
- Does the proteomic experiment quantify the level of phosphor-protein, or total proteins? I ask because Fig. 4a legend says "Workflow for TMT-labeled phosphopeptides in WT, RC, and GS striatal crude synaptosomal fraction...", suggesting that only phosphopeptides were labelled. If this was the case, 1) was any steps taken to enrich for phosphoproteins during protein extraction? 2) It probably should be stated more clearly in figures when protein comparisons between genotypes involves phosphor-protein or total proteins.
- Fig. 4c, the labels for the Venn diagram looks strange. Do the authors mean that the red circle is number of proteins changed for WT vs. RC and the blue circle is number of proteins changed for WT vs GS?
- Results, "Increased synaptic PKA activities in the striatum of R1441C mice following striatal

motor learning" section, "we used a pan phospho-PKA substrate antibody to detect phosphorylation of downstream PKA targets...". I could not find any mention of the anti-phospho-PKA substrate antibody used in the Methods section. This should be added.

- Suppl Fig. 2b: "The rheobase in RC mice was increased for both dSPNs (rheobase_{WT} = 250 ± 25, n = 13 cells; rheobase_{RC} = 325 ± 50, n = 29 cells; p = 0.0078, Mann-Whitney U test) and iSPNs (rheobase_{WT} = 162.5 ± 87, n = 20 cells; rheobase_{RC} = 225 ± 50, n = 28 cells; p = 0.0046, Mann-Whitney U test)." The comparison for 2b is for GS and not for RC, according to the figure.

- Suppl Fig. 6b legend "b. Representative WB analysis of wt, RC, and GS P2 fractions probed for LRRK2, D1R, D2R, DAT, ..." the heading "b" in the figure is missing for the western blot, and instead is used for the bar charts.

- Suppl Fig. 6b: do the authors have data showing that the synaptosomal fraction is enriched for synaptic marker (e.g., PSD95) relative to to S1 fraction or the P1 fraction?

Reviewer #3 (Remarks to the Author):

I congratulate with the authors for the thorough revisions and the quality of their work. One last note: in fig. 4C the Venn diagram for the WT vs. RC comparison is missing

Reviewer #4 (Remarks to the Author):

The authors have addressed all our previous concerns and questions.

Response to Reviewers

We want to thank the reviewers again for their very helpful feedback on our work. We greatly appreciate the comments that helped further clarify or expand our discussion. We have taken into account the points raised and have also corrected some of the errors in the figures. Additionally, to fully address the Reviewer's comments, we have included a new Supplemental Figure. Our point-by-point responses to the reviewers' comments are below. The relevant changes in the text are shown in red.

Reviewer #1 (Remarks to the Author):

The manuscript has been extensively revised with addition of several new data. Notably the authors now provide data on PKA activity increase in RC mutant and show potential mechanisms of reduced excitability in SPN as opposed to GS mutant with similar DA release changes without excitability change. Overall the manuscript provides very interesting and in-depth insights on the effect of LRRK2 mutations on striatal function.

- Results, "Increased synaptic PKA activities in the striatum of R1441C mice following striatal motor learning" section, "The phospho-PKA levels in the striatal synaptosomal preparations were decreased at the end of the task in the WT mice ; at a time point, their initial impaired performance was improved, similar to their saline controls. However, the PKA signaling remained elevated in the RC mice correlating with their impaired performance in the rotarod test". Although the western blot in Fig. 4g seems to agree with this claim, there was no quantification or statistics. These should be included to substantiate the claim that p-PKA substrates decreased for WT from session 5 to 18, but remained both high across time points, and higher than WT, for RC.

Response 1.1. We have now quantified the Western Blot (WB) in **Figure 4g**. The quantification and statistical details of the one-way ANOVA test performed are now shown in **Suppl. Fig 8a** and detailed in the Results section, under "Increased synaptic PKA activities in the striatum of R1441C mice following striatal motor learning."

- Results, "Aberrant PKA signaling underlies the motor deficits of R1441C mice". Although sh-PKARIIB increased performance of RC mice previously treated with DA receptor antagonist, it had the opposite effects on WT mice and reduced their performance. Could the authors comment on this counterintuitive finding? For instance, does it imply that motor performance isn't a simple monotonic function of p-PKA substrate? Could PKARIIB knockdown in dSPN vs. iSPN play a role in this counterintuitive finding?

Response 1.2. Our data does support that motor performance is not a monotonic function of p-PKA by showing motor learning is sensitive to the level of PKA activity. Specifically, in RC mice with elevated PKA activity levels, sh-PKARIIB improved performance, whereas in WT mice with normal PKA activity, sh-PKARIIB reduced performance. In addition, the non-monotonic function of PKA has been reported by others who showed that learning is dependent upon the activity of PKA in dSPNs and iSPNs, which are dichotomously sensitive to increases or decreases in dopamine, respectively (PMID: 33361810). In the context of the technical limitations of our study, we discussed the outstanding question remaining whether, in RC mice, dSPNs versus iSPNs undergo differential regulation of PKA activity. However, our voltammetric and electrophysiological data showed that in RC mice dopamine is reduced and iSPNs become hypoexcitable. The reduction of dopamine and hypoexcitability of iSPNs in RC mice further suggests that a specific

dysfunction of RC mutation exists in iSPNs and not dSPNs or PKA activity in RC iSPNs are more sensitive and dysregulated with reduced dopamine. More targeted experiments in identified mutant LRRK2 iSPNs and dSPNs are needed to delineate how LRRK2 mutations shape PKA signaling. Relatedly, it is important to note that a coordinated and concurrent activation of direct and indirect pathways is needed for normal motor output (PMID:23354054) and that disruption of iSPNs leads to a functional imbalance of basal ganglia circuitry (PMID:24411738). We have now integrated these important points into our revised discussion and results section (See **Response 1.3**).

- Results, "Aberrant PKA signaling underlies the motor deficits of R1441C mice". In addition to the single western blot shown in Fig. 5b, do the authors have quantification or statistics of PKARII β or p-PKA substrate levels after sh-control vs. sh-PKARII β ? For instance, did sh-PKARII β reduce p-PKA substrate in RC mice back to WT's sh-control level, thereby rescuing motor performance? Additionally, did sh-PKARII β reduce WT's p-PKA substrate to an even lower level (e.g., reference 73, Brandon et al. 1998)? Data like these would suggest that there may be an optimal level of p-PKA substrate for motor performance.

Response 1.3. We agree with the Reviewer that adding these data will strengthen the connection between PKA and striatal motor learning. Thus, in our revised manuscript, we have quantified the PKARII β levels after the viral constructs injection to knock down the PKARII β gene, compared to sh-control injected mice. This is now shown in **Suppl Fig. 8b-c**. In addition, we quantified the effect of PKARII β knockdown in the PKA signaling as evidenced by the levels of the p-PKA substrate in the striatal extracts of sh-control and sh-PKARII β WT and RC mice. While the effect of PKARII β knockdown was similar across genotypes in regards to PKARII β protein expression, we observed interesting alterations of PKA signaling as evidenced by p-PKA substrates under these conditions. Specifically, our data show that sh-PKARII β administration in RC mice returned the abnormally elevated PKA signaling of these mice to levels similar to control levels (WT sh-control). Moreover, the sh-PKARII β decreased the levels in the WT mice compared to WT sh-control as well as RC mice injected with sh-PKARII β . Given that these alterations in PKA activities are followed by differences in motor performance (**Figure 5** and **Response 1.2**), our findings suggest that too much or too little PKA signaling impairs motor performance. We have included this description in the Results and Discussion sections.

Minor Comments:

- Methods, "Viral-mediated short-hairpin RNA knockdown" section: when was the shRNA AAV injected relative to the rotarod experiment?

Response 1.4. The shRNA AAVs were injected four weeks before the rotarod experiment. This information is now added in the Methods section.

- Methods, "Quantitative proteomics and analysis" section, "129 for WT, 130 for RC, and 131 for GS for three biological replicates per the manufacturer's instructions". Please clarify: does this mean 3 biological replicates for each genotype, or 3 replicates in total (1 for each genotype)?

Response 1.5. There are 3 biological replicates for each genotype. We mixed one of each (one WT, one RC, and one GS) into one tube before performing TMT-LC/MS. We ran three tubes (3 WT, 3 RC, and 3 GS) for TMT-LC/MS.

• Fig. 3: there does not seem to be any mention of Fig. 3e in the main text. -What is its purpose? And what is the inset figure in Fig. 3e and its purpose?

Response 1.6. The paragraph describing the open field effect and corresponding to **Figure 3e** was accidentally deleted between the original and revised manuscript. We have now included it in the Results section.

Regarding the inset, it shows that acute D1 and D2 receptor antagonism, 2 hours prior to open field, substantially reduced open-field activity. This is in contrast with no difference in the total distance traveled when a subset of mice after rotarod training (**Figure 3a**) and D1 and D2 antagonists administration were tested in the open field test following the 72-hr break (**Figure 3e**). These findings suggest that despite the acute effect on motor performance, the dopamine receptor antagonism leading to an impeded experience-dependent learning impairs specifically the subsequent striatal motor learning performance/recovery even after the restoration of dopamine signaling.

Results: "R1441C mice show impaired dopamine-dependent motor learning" section, "D1 receptor antagonism immediately improved the drug-free phase in all genotypes". The sentence seems potentially confusing? Maybe "Mice previously treated with D1 receptor antagonist showed immediate rotarod improvement after 72 hr washout, regardless of genotype"?

Response 1.7. We thank the Reviewer for the suggestion, and we have now changed this sentence based on his/her suggestion.

• For synaptosome fraction extraction, both for western blot and proteomic analysis (Fig. 4 and Suppl Fig. 6), there is no information about when the striatum was extracted relative to rotarod training and dopamine receptor antagonist. Was it immediately after 5 sessions of rotarod training with dopamine receptor antagonist treatments? Or was it after the 72 hour washout period? Or some other time point? This can be important because dopamine antagonist itself can have time-dependent effect on the proteome, especially on protein phosphorylation.

Response 1.8. We euthanized the mice and collected tissue 24hrs after the 5th session. The tissue was kept frozen at -80°C . We have added this description in the Methods section.

• Does the proteomic experiment quantify the level of phosphor-protein, or total proteins? I ask because Fig. 4a legend says "Workflow for TMT-labeled phosphopeptides in WT, RC, and GS striatal crude synaptosomal fraction...", suggesting that only phosphopeptides were labelled. If this was the case, 1) was any steps taken to enrich for phosphoproteins during protein extraction? 2) It probably should be stated more clearly in figures when protein comparisons between genotypes involves phosphor-protein or total proteins.

Response 1.9. Our analysis was a proteomic screening for total proteins. The **Figure 4a** legend reflects that now.

• Fig. 4c, the labels for the Venn diagram looks strange. Do the authors mean that the red circle is number of proteins changed for WT vs. RC and the blue circle is number of proteins changed for WT vs GS?

Response 1.10. The red circle shows the number of proteins changed in WT vs. RC comparison and the blue circle indicates the number of proteins changed in WT vs. GS P2 striatal extracts. We have updated the labels to demonstrate this more clearly.

- Results, "Increased synaptic PKA activities in the striatum of R1441C mice following striatal motor learning" section, "we used a pan phospho-PKA substrate antibody to detect phosphorylation of downstream PKA targets...". I could not find any mention of the anti-phospho-PKA substrate antibody used in the Methods section. This should be added.

Response 1.11. The phospho-PKA substrate antibody information is now included in the Methods section.

- Suppl Fig. 2b: "The rheobase in RC mice was increased for both dSPNs (rheobase_{WT} = 250 ± 25, n = 13 cells; rheobase_{RC} = 325 ± 50, n = 29 cells; p = 0.0078, Mann–Whitney U test) and iSPNs (rheobase_{WT} = 162.5 ± 87, n = 20 cells; rheobase_{RC} = 225 ± 50, n = 28 cells; p = 0.0046, Mann–Whitney U test)." The comparison for 2b is for GS and not for RC, according to the figure.

Response 1.12. We thank the Reviewer for pointing out this error. It has been corrected in the legend.

- Suppl Fig. 6b legend "b. Representative WB analysis of wt, RC, and GS P2 fractions probed for LRRK2, D1R, D2R, DAT, ..." the heading "b" in the figure is missing for the western blot, and instead is used for the bar charts.

Response 1.13. We have now corrected the labeling of Figure panels.

- Suppl Fig. 6b: do the authors have data showing that the synaptosomal fraction is enriched for synaptic marker (e.g., PSD95) relative to to S1 fraction or the P1 fraction?

Response 1.14. We have now included a WB showing enrichment of PSD95 in the P2 fraction (**Suppl. Fig.6a**). Our recent previous publication consists of a detailed description of our subcellular fractionation protocol showing the successful enrichment of PSD95 in the synaptosomal fractions. Additionally, quantitative analysis revealed no difference in relative levels of PSD-95 protein in the PSD fraction across genotypes (PMID: 33006315).

Reviewer #3 (Remarks to the Author):

I congratulate with the authors for the thorough revisions and the quality of their work.

One last note: in fig. 4C the Venn diagram for the WT vs. RC comparison is missing

Response 3.1. We thank the Reviewer for their help in strengthening this work and for pointing out this error. We have updated the labels to reflect the correct comparisons (See **Response 1.10**).

Reviewer #4 (Remarks to the Author):

The authors have addressed all our previous concerns and questions.

Response 4.1. We greatly thank the Reviewer for their helpful feedback in strengthening our work.

REVIEWERS' COMMENTS:

Reviewer #5 (Remarks to the Author):

As I haven't reviewed the paper in first two rounds (and I don't have access to previous submissions) I will only comment on the points raised by referee 1 in round 2 and on the authors' point-to-point reply. Reviewers 3 and 4 are happy with this revision, but reviewer 1 (R1) still had a few concerns.

R1 asked for quantification of Western Blot (WB) data that are now provided in the supplement. I am not sure why the authors decided to put the data in the supplement, but at least the data are given as requested.

Secondly, R1 enquired about the aberrant PKA signalling. The authors have now provided a quantitative description of data as requested (e.g. Suppl. Fig. 8). Again, I don't know why this data is only shown in the supplement, I would have liked to see it in the main manuscript.

All minor issues have been sufficiently addressed.

There is a typo in line 344: I guess it should read 'Sch23390'.

Response to Reviewers

We want to thank the reviewers again for their constructive feedback on our work along the way. We are pleased that our responses fully addressed the Reviewers' comments. Our point-by-point responses to the final Reviewer's 'comments are below. The relevant changes in the text are shown in red.

REVIEWERS' COMMENTS:

Reviewer #5 (Remarks to the Author):

As I haven't reviewed the paper in first two rounds (and I don't have access to previous submissions) I will only comment on the points raised by referee 1 in round 2 and on the authors' point-to-point reply. Reviewers 3 and 4 are happy with this revision, but reviewer 1 (R1) still had a few concerns.

R1 asked for quantification of Western Blot (WB) data that are now provided in the supplement. I am not sure why the authors decided to put the data in the supplement, but at least the data are given as requested.

Response: Following R5 suggestion, the quantification data of WB are now part of the main Figure 4 (Figure 4h).

Secondly, R1 enquired about the aberrant PKA signalling. The authors have now provided a quantitative description of data as requested (e.g. Suppl. Fig. 8). Again, I don't know why this data is only shown in the supplement, I would have liked to see it in the main manuscript

Response: The representative WB analysis data and the quantification of PKA signaling has now been moved to Figures 6a,b.

All minor issues have been sufficiently addressed.

There is a typo in line 344: I guess it should read 'Sch23390'.

Response: This typo has now been corrected.